# Augmented Invertible Koopman Autoencoder
# for long-term time series forecasting

**Anthony Frion**                                                      *anthony.frion@hereon.de*
*Institute of Coastal Systems - Analysis and Modeling*
*Helmholtz-Zentrum Hereon, Geesthacht, Germany*

**Lucas Drumetz**                                                    *lucas.drumetz@imt-atlantique.fr*
*IMT Atlantique*
*Lab-STICC, UMR CNRS 6285, Brest, France*

**Mauro Dalla Mura**                               *mauro.dalla-mura@gipsa-lab.grenoble-inp.fr*
*Université Grenobles Alpes*
*Grenoble INP*
*GIPSA-lab, Grenoble, France*
*Institut Universitaire de France*

**Guillaume Tochon**                                              *guillaume.tochon@lrde.epita.fr*
*LRE EPITA, Le Kremlin-Bicêtre, France*

**Abdeldjalil Aïssa El Bey**                              *abdeldjalil.aissaelbey@imt-atlantique.fr*
*IMT Atlantique*
*Lab-STICC, UMR CNRS 6285, Brest, France*

**Reviewed on OpenReview:** *https://openreview.net/forum?id=o6ukhJLzMQ*

## Abstract

Following the introduction of Dynamic Mode Decomposition and its numerous extensions, many neural autoencoder-based implementations of the Koopman operator have recently been proposed. This class of methods appears to be of interest for modeling dynamical systems, either through direct long-term prediction of the evolution of the state or as a powerful embedding for downstream methods. In particular, a recent line of work has developed invertible Koopman autoencoders (IKAEs), which provide an exact reconstruction of the input state thanks to their analytically invertible encoder, based on coupling layer normalizing flow models. We identify that the conservation of the dimension imposed by the normalizing flows is a limitation for the IKAE models, and thus we propose to augment the latent state with a second, non-invertible encoder network. This results in our new model: the Augmented Invertible Koopman AutoEncoder (AIKAE). We demonstrate the relevance of the AIKAE through a series of long-term time series forecasting experiments, on satellite image time series as well as on a benchmark involving predictions based on a large lookback window of observations.

## 1 Introduction

A longstanding question in dynamical systems theory has been the ability to characterize the behavior of dynamical systems from which one does not have access to the equations that govern their evolution, but only to data snapshots measured from them. With the increasing computational resources and the development of autodifferentiation frameworks, data-driven methods, and specifically deep neural networks, have taken an increasing importance in dynamical systems modeling.

Among these neural network methods, an increasing part has been designed based on Koopman operator theory, which means that they seek to find a representation of the state from which the evolution through time can be described linearly. A popular class of such models is the Koopman autoencoder (Lusch et al., 2018), which simply consists of a neural autoencoder along with a matrix that describes the linear dynamics in the latent space of the encoder. Many flavors of the Koopman autoencoder seek to improve the long-term stability of the linear latent dynamics through constrained parameterizations of its governing matrix (Bevanda et al., 2022; Fan et al., 2022; Zhang et al., 2024) or additional loss function terms (Azencot et al., 2020; Frion et al., 2023a). Some works propose to use the representation learned by a Koopman autoencoder in a broader computation pipeline, for example as embeddings for a Transformer model (Geneva & Zabaras, 2022; Jin et al., 2023) or in a data assimilation framework (Frion et al., 2024; Singh et al., 2024). In the present work, we take interest in recent advancements (Meng et al., 2024; Jin et al., 2023) consisting in implementing the Koopman autoencoder with a coupling layer normalizing flow as the encoder and the analytical inverse of this flow as the decoder. We show that the induced constraint on the dimension of the latent space is detrimental to the ability of the model to find a Koopman invariant subspace. As a remedy, we propose to learn a second encoder in order to inflate the latent dimension of the model, without changing the architecture of the decoder. The resulting model is our Augmented Invertible Koopman AutoEncoder (AIKAE).

We perform long-term forecasting experiments with this model in two settings. First, we work on regularly-sampled time series with no missing observations, where one has access to numerous past observations in order to compute a forecast. For this setting, we show that a delayed AIKAE, i.e. a AIKAE from which the input space contains multiple consecutive observations rather than a single one, can obtain accurate results, challenging a set of strong and recent baselines. Then, we work on satellite image time series, where there are usually a lot of missing observations, resulting in irregularly-sampled data. In this context, we use a pre-trained AIKAE model as a dynamical prior in a constrained variational data assimilation framework. We show that this model performs better in this task than other Koopman autoencoder variants. The codes associated to our experiments are available at `https://github.com/anthony-frion/AIKAE`.

The remainder of this paper is organized as follows: in section 2, we review Koopman operator theory and the recent related neural network-based models. In section 3, we introduce our new architectures, including the AIKAE architecture in subsection 3.1 and the delayed Koopman autoencoders in subsection 3.2. In section 4, we show how an AIKAE can be used as a dynamical prior in a variational data assimilation cost. Our experiments on long-term forecasting with a fixed lookback window and on assimilating satellite image time series are respectively presented in sections 5 and 6. Section 7 concludes our work.

## 2 Background and related works

Originally introduced in Koopman (1931), Koopman operator theory has known a renewed interest in the last few decades, starting from the work of Mezić (2005). We refer the interested reader to Brunton et al. (2022) for an extensive review of Koopman operator theory and its applications. In a few words, this theory states that any dynamical system, regardless of its inherent complexity, can be described by a linear operator, although at the cost of an infinite dimension in the general case. More precisely, let us introduce a (supposedly autonomous and deterministic) dynamical system from which the state at a given time can be described by a $n$-dimensional variable $\mathbf{x} \in \mathcal{X} \subset \mathbb{R}^n$. The system is defined by a discrete-time evolution operator $F : \mathcal{X} \to \mathcal{X}$. Then, assuming that the state of the system at an integer time $t$ is $\mathbf{x}_t \in \mathcal{X}$, we define

$$\mathbf{x}_{t+1} \triangleq F(\mathbf{x}_t). \tag{1}$$

The Koopman operator $\mathcal{K}$ is such that, for any measurement function $g : \mathcal{X} \to \mathbb{R}$ and initial condition $\mathbf{x}_t$,

$$\mathcal{K}g(\mathbf{x}_t) \triangleq (g \circ F)(\mathbf{x}_t) = g(\mathbf{x}_{t+1}). \tag{2}$$

Thus, in theory, one would simply need to have access to the expression of $\mathcal{K}$ for the canonical measurement functions (i.e. the projections of the full state $\mathbf{x}$ onto its $n$ variables) to be able to exactly characterize the dynamical system $F$. However, the infinite dimension of the space of measurement functions means that the Koopman operator is itself infinite dimensional, and therefore often difficult to describe in practice.

For this reason, most of the data-driven methods inspired by the Koopman operator consist in finding an approximation of this operator on a specific $d$-dimensional set of measurement functions $(g_1, ..., g_d)$. Ideally, one would require this set to be invariant by the Koopman operator. This would mean that, for any of the functions $g_i$, there would exist coefficients $k_{i,.}$ such that, for any initial condition $\mathbf{x}_t \in \mathcal{X}$,

$$\mathcal{K}g_i(\mathbf{x}_t) = g_i(F(\mathbf{x}_t)) = \sum_{j=1}^{d} k_{i,j} g_j(\mathbf{x}_t). \tag{3}$$

In this case, the action of the Koopman operator on the subspace spanned by $(g_1, ..., g_d)$ could be simply described by a matrix $\mathbf{K} \in \mathbb{R}^{d \times d}$, built with the coefficients $k_{i,j}$. For linear dynamical systems, the space spanned by the canonical measurement functions of the system (i.e. the functions constituting the state variables) is obviously invariant by the Koopman operator. For nonlinear dynamical systems, there are some cases in which a finite-dimensional Koopman invariant subspace containing all the state variables (in addition to some "augmentation" variables, required to obtain the linearity) are known. Examples of such dynamical systems are detailed in e.g. Brunton et al. (2016) and Kutz et al. (2016). However, most of the time the Koopman invariance has to be approximated to a certain degree, since in the general case there exists no finite-dimensional Koopman invariant subspace for the system. Once a set of measurement functions $(g_1, ..., g_d)$ has been designed, one typically looks for the matrix $\mathbf{K}^*$ that minimizes the residual error of the multiplication by a matrix $\mathbf{K}$. Formally, one can work with a set of data $\mathbf{X} = (\mathbf{x}_1, ..., \mathbf{x}_T)$, with a time-shifted version $\mathbf{Y} = (\mathbf{x}_1', ..., \mathbf{x}_T')$, where, for any index $1 \leq t \leq T$, $\mathbf{x}_t' = F(\mathbf{x}_t)$. Then, we seek to find

$$\mathbf{K}^* = \underset{\mathbf{K} \in \mathbb{R}^{d \times d}}{\arg \min} ||\mathbf{K}\mathbf{g}(\mathbf{X}) - \mathbf{g}(\mathbf{Y})||^2, \tag{4}$$

where we use the notations

$$\mathbf{g}(\mathbf{X}) = \begin{pmatrix} g_1(\mathbf{x}_1) & g_1(\mathbf{x}_2) & & g_1(\mathbf{x}_T) \\ \vdots & \vdots & \cdots & \vdots \\ g_d(\mathbf{x}_1) & g_d(\mathbf{x}_2) & & g_d(\mathbf{x}_T) \end{pmatrix}, \quad \mathbf{g}(\mathbf{Y}) = \begin{pmatrix} g_1(\mathbf{x}_1') & g_1(\mathbf{x}_2') & & g_1(\mathbf{x}_T') \\ \vdots & \vdots & \cdots & \vdots \\ g_d(\mathbf{x}_1') & g_d(\mathbf{x}_2') & & g_d(\mathbf{x}_T') \end{pmatrix}. \tag{5}$$

The optimization problem of equation 4 can be solved using the well-known least-squares solution:

$$\mathbf{K}^* = \mathbf{g}(\mathbf{Y})(\mathbf{g}(\mathbf{X}))^+, \tag{6}$$

where $\cdot^+$ denotes the Moore–Penrose pseudoinverse. It should be noted that this solution only accounts for the advancement of one time step, i.e. one iteration of the discrete dynamics $F$. Hence, the obtained model will generally perform poorly in long-term predictions. For this reason, while early Koopman-based methods such as dynamic mode decomposition (Schmid, 2010) and extended dynamic mode decomposition (Williams et al., 2015) compute the least-square solution of equation 6 (or a low-rank approximation of it), subsequent neural network-based implementations generally leverage trajectories with multiple time steps in order to train a model that produces accurate long-term predictions.

To sum up, many practical implementations of the Koopman operator consist in finding a set $\mathbf{g} : \mathcal{X} \to \mathbb{R}^d$ of $d$ measurement functions $(g_1, ..., g_d)$, each from the state space $\mathcal{X}$ to $\mathbb{R}$, and a matrix $\mathbf{K} \in \mathbb{R}^{d \times d}$ that approximates the restriction of the Koopman operator to the subspace spanned by these functions. We have mentioned the importance of choosing a set of measurement functions that span an (approximately) Koopman invariant subspace. Another important aspect of these methods is the ability to faithfully reconstruct an input state $\mathbf{x} \in \mathcal{X}$ from its embedding $\mathbf{g}(\mathbf{x}) \in \mathbb{R}^d$, and to do the same for the time-advanced embeddings $\mathbf{K}\mathbf{g}(\mathbf{x})$, in order to produce predictions for the evolution of the state vector from any initial condition. Thus, one must be able to define a (possibly approximated) function $\mathbf{f} : \mathbb{R}^d \to \mathcal{X}$ such that the composition $\mathbf{f} \circ \mathbf{g}$ is (approximately) equal to the identity function. The reconstruction abilities of the recently introduced classes of Koopman-based methods are discussed in Jin et al. (2024). We defer a detailed discussion of older methods to appendix A and directly discuss Koopman autoencoders (KAEs), a class of methods introduced by Lusch et al. (2018) and extended by numerous subsequent works, e.g. Otto & Rowley (2019); Li et al. (2020); Azencot et al. (2020); Berman et al. (2023); Frion et al. (2024) among many others. These methods model the Koopman invariant subspace through the means of a neural autoencoder, which does not directly

include the state variables. A neural autoencoder simply consists of two neural networks, $\phi$ and $\psi$, each with its set of trainable parameters, from which the composition is approximately equal to the identity function, i.e. $\psi \circ \phi(\mathbf{x}) \approx \mathbf{x}$. For KAE models, the encoder $\phi : \mathbb{R}^n \to \mathbb{R}^d$ learns a non-trivial Koopman invariant set of measurement functions, while the decoder $\psi : \mathbb{R}^d \to \mathbb{R}^n$ learns to reconstruct the state space from the latent representation of $\phi$. Depending on the implementations, the Koopman matrix $\mathbf{K} \in \mathbb{R}^{d \times d}$ is learned alongside the parameters of $\phi$ and $\psi$ or obtained separately through the resolution of a least squares problem as in equation 4. The predictions of a KAE model after $\tau$ time steps are computed as:

$$\mathbf{x}_{t+\tau} \approx \hat{\mathbf{x}}_{t+\tau} = \psi(\mathbf{K}^\tau \phi(\mathbf{x}_t)). \tag{7}$$

While general neural autoencoder models were originally introduced for the purpose of reducing the dimension of the input $\mathbf{x}$ (i.e. following the property $d < n$), in the context of finding a better representation of the Koopman operator, it might actually be beneficial to learn a latent representation with a higher dimension than the input (i.e. $d > n$). In practice, the latent dimension $d$ should be regarded as an important parameter for the design of a KAE model.

Although the Koopman autoencoder framework enables for a high expressivity in the search of a suitable Koopman invariant subspace, it has the notable inconvenience that, in contrast to earlier methods, it computes an approximated rather than an exact reconstruction of the input state. In practice, in the loss function for training a Koopman autoencoder, one should include a reconstruction term to ensure that the components $\phi$ and $\psi$ indeed constitute an autoencoder. This loss function term is to be minimized in conjunction with the prediction loss term and to the linearity loss term (see e.g. Lusch et al. (2018)), which leads to a complex loss landscape, and possibly to difficulties in adjusting the relative weights of the loss function terms. For this reason, a recent line of work (Jin et al., 2023; Meng et al., 2024; Jin et al., 2024) has investigated the substitution of the neural autoencoder $(\phi, \psi)$, by an analytically invertible $\phi$ with its exact inverse $\phi^{-1}$. In this case, the predictions of the model are given by:

$$\mathbf{x}_{t+\tau} \approx \hat{\mathbf{x}}_{t+\tau} = \phi^{-1}(\mathbf{K}^\tau \phi(\mathbf{x}_t)). \tag{8}$$

This results in a subclass of methods which we call invertible Koopman autoencoders (IKAEs). More specifically, they proposed to implement $\phi$ with coupling-layer normalizing flow models (Dinh et al., 2014; 2017; Kingma & Dhariwal, 2018). These models have several interesting properties. Notably, they indeed have an analytical inverse transformation, enabling an (algebraically) exact reconstruction[1] of an encoded state $\mathbf{x}$. In addition, their Jacobians are tractably computable, which gives them a potential for stochastic modeling. We provide more background on this and on normalizing flows in appendix B.

Another important property of coupling-layer normalizing flows is that they always preserve the dimension of the input state, which is a necessity in order for the change of variable formula to be applicable. This may be detrimental for IKAE models since, as previously mentioned, one often needs to inflate the dimension of the state space in order to obtain a good approximation of the Koopman operator. To alleviate this issue, the authors of the existing IKAE models have proposed to concatenate zeros to the state vector, either before (Meng et al., 2024) or in-between (Jin et al., 2023) the coupling layers of the normalizing flow. However, this approach means that the resulting model will learn a function that enables to reconstruct these added zeros by design, while only the reconstruction of the true state variables is of interest. In addition, the operation of concatenating zeros to the state vector prevents a direct application of the change-of-variable formula from equation 19, hence reducing the possibilities for stochastic extensions.

## 3 Our proposed Koopman autoencoder architectures

### 3.1 Augmented invertible Koopman autoencoder

We mentioned in the previous section that the base IKAE architecture had the inconvenience of not enabling to learn a large enough set of measurement functions to obtain a sufficiently good approximation of the

---

[1]It should be noted that the algebraic invertibility does not guarantee stable and accurate reconstructions in practice, since normalizing flows can be subject to numerical ill-conditioning: see e.g. Lee et al. (2021) for an extensive discussion on the conditioning of normalizing flow models.

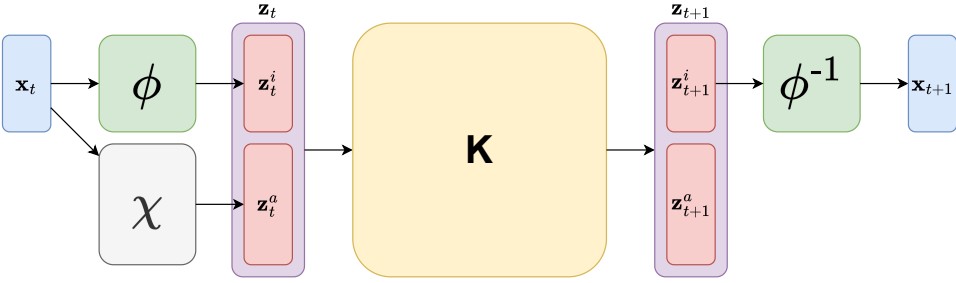

Figure 1: Graphical representation of the AIKAE architecture. $\phi$ is a coupling layer normalizing flow with an analytical inverse $\phi^{-1}$, $\chi$ is a neural network (generally a simple multi-layer perceptron in practice) and $\mathbf{K}$ is a matrix.

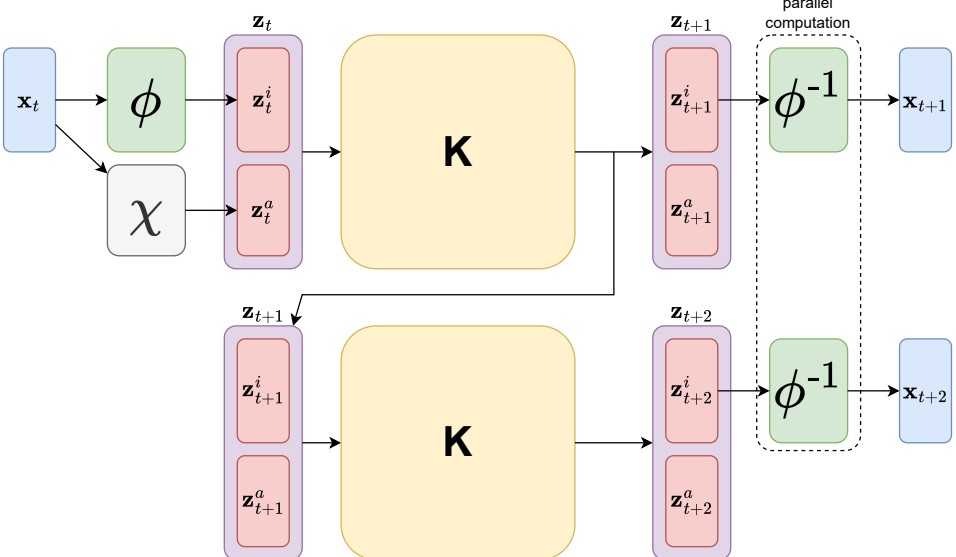

Figure 2: Graphical representation of the prediction over 2 future timesteps with an AIKAE model. This can be easily generalized to long-term predictions. As can be seen, the forecasting computations are all performed in the latent space, and the predicted latent states can all be decoded in parallel for more efficiency.

Koopman operator. Thus, in this section, we propose a revised architecture in which the invertible encoder of these models is augmented with a second neural encoder $\chi$, which enables to capture a richer set of measurement functions while keeping the invertibility of the model. We call the resulting architecture an augmented invertible Koopman autoencoder (AIKAE). This architecture is represented graphically in figure 1. Additionally, we represent the prediction over 2 time steps in figure 2, and the generalization to an arbitrarily long $T$ time steps prediction follows directly from it. Since $\chi$ does not have to be invertible, it can be implemented with any neural network architecture.

The predictions performed by the AIKAE model over $\tau$ time steps from an observed initial condition $\mathbf{x}_t$ can be described as follows:

$$\mathbf{z}_t = \begin{pmatrix} \mathbf{z}_t^i \\ \mathbf{z}_t^a \end{pmatrix} = \begin{pmatrix} \phi(\mathbf{x}_t) \\ \chi(\mathbf{x}_t) \end{pmatrix} \tag{9}$$

$$\mathbf{z}_{t+\tau} = \begin{pmatrix} \mathbf{z}_{t+\tau}^i \\ \mathbf{z}_{t+\tau}^a \end{pmatrix} = \mathbf{K}^\tau \mathbf{z}_t \tag{10}$$

$$\hat{\mathbf{x}}_{t+\tau} = \phi^{-1}(\mathbf{z}_{t+\tau}^i) \tag{11}$$

The innovation of the AIKAE model in comparison to the IKAE model is that we introduce a second encoder $\chi : \mathbb{R}^n \to \mathbb{R}^p$, which we call the augmentation encoder. The latent state $\mathbf{z}_t$ corresponding to $\mathbf{x}_t$ is thus obtained by concatenating an augmentation encoding $\mathbf{z}_t^a = \chi(\mathbf{x}_t)$ to the invertible encoding $\mathbf{z}_t^i = \phi(\mathbf{x}_t)$ produced by the unchanged normalizing flow model $\phi : \mathbb{R}^n \to \mathbb{R}^n$, as summarized in equation 9. Then, the linear latent dynamics is defined by the multiplication of the full latent state $\mathbf{z}_t$ by the matrix $\mathbf{K}$, which is now of size $d = n + p$, as summarized in equation 10. Hence, by adding an augmentation part to the encoding, we indeed increase the number of measurement functions included in the latent space, and the dimension of the approximated Koopman operator $\mathbf{K}$ as a consequence. Finally, in order to go back to the input space after any desired number $\tau$ of iterations, one can decode the invertible part $\mathbf{z}_{t+\tau}^i$ of $\mathbf{z}_{t+\tau}$, as shown in equation 11. Note that, with our notations, the operations $\mathbf{z}^i$ and $\mathbf{z}^a$ respectively correspond to projections on the first $n$ or the last $p$ variables of $\mathbf{z} \in \mathbb{R}^{n+p}$.

From studying these equations, one can see that the augmentation part $\mathbf{z}_t^a$ of the initial latent state has no influence on the direct reconstruction $\hat{\mathbf{x}}_t$ (i.e. the case where $\tau = 0$), which is still algebraically exact thanks to the analytical inverse $\phi^{-1}$ of $\phi$. However, as evidenced by equation 10, $\mathbf{z}_t^a$ has an impact on the subsequent invertible parts of the encoding through the multiplications by $\mathbf{K}$ if and only if $\mathbf{z}_t^a$ is not in the nullspace of the upper-right block of $\mathbf{K}$. An immediate corollary of this observation is that the upper-right block of $\mathbf{K}$ should be nonzero in order for some information to flow from $\mathbf{z}_t^a$ to $\mathbf{z}_{t+1}^i$ and subsequent invertible encodings. In fact, should the last $p$ columns of $\mathbf{K}$ be zero, then the augmentation part $\mathbf{z}_t^a$ would have no influence on the predictions in the state space, which means that the whole model would be equivalent to a non-augmented IKAE model. Thus, we have the intuitive result that the AIKAE architecture is a generalization of the IKAE architecture. In addition, one may interpret the invertible and augmentation parts of an encoding $\mathbf{z}_t$ as a disentanglement between the "static features" and the "dynamical features". This interpretation is particularly interesting when performing data assimilation using the methods of section 4. In practice, the output size $p$ of $\chi$ determines the latent size $d = n+p$ of an AIKAE, making it an important hyperparameter, similarly to the latent dimension $d$ itself for non-invertible KAE models.

In order to characterize the predictions by an AIKAE in a more compact way, we introduce the global encoder $\Phi : \mathbb{R}^n \to \mathbb{R}^d$ which corresponds to the concatenation of the invertible encoder $\phi$ and the augmentation encoder $\chi$, i.e. $\mathbf{z}_t = \Phi(\mathbf{x}_t)$. Correspondingly, we have that the global decoder $\Phi^{-1} : \mathbb{R}^d \to \mathbb{R}^n$ consists in the application of $\phi^{-1}$ on the invertible part (i.e. the first $n$ variables) of a latent vector. Thus, equations 9 to 11 can now be summarized as

$$\hat{\mathbf{x}}_{t+\tau} = \Phi^{-1}(\mathbf{K}^\tau \Phi(\mathbf{x}_t)). \tag{12}$$

As the notations suggest, $\Phi^{-1}$ is still an analytical left inverse of $\Phi$, as $\Phi^{-1} \circ \Phi$ corresponds to the identity function. One should however be aware that the reversed composition $\Phi \circ \Phi^{-1}$ is not an identity function since the information on the augmentation part of the encoding is dismissed when computing $\Phi^{-1}$. Thus, $\Phi^{-1}$ is not a right inverse of $\Phi$.

## 3.2 Delayed Koopman autoencoders

We now discuss delayed Koopman autoencoders, which simply consist in KAE models that take as their input state a concatenation of $m$ consecutive observed states from a dynamical system rather than a state vector corresponding to a specific time index. We refer to this approach as a delay embedding. Formally, when observing a long time series $(\mathbf{x}_0, ..., \mathbf{x}_T) \in \mathcal{X}^{T+1}$, rather than directly using a state $\mathbf{x}_t \in \mathbb{R}^n$ as the input to a KAE, one may alternatively use

$$\mathbf{y}_t = \begin{pmatrix} \mathbf{x}_{tm} \\ \vdots \\ \mathbf{x}_{tm+m-1} \end{pmatrix} \in \mathbb{R}^{n'} \tag{13}$$

as the input to the model. Then, the dimension of the input space of the model will be $n' = nm$. Thus, in order to avoid manipulating a very high-dimensional input state (taking into account the fact that the

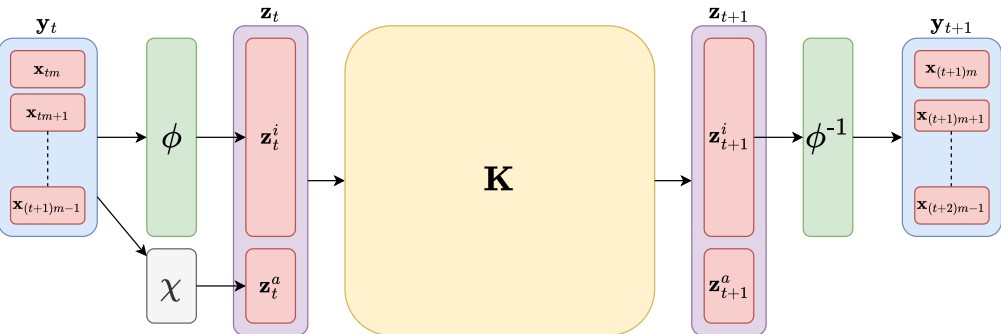

Figure 3: Graphical representation of an AIKAE model with delay embedding of the state. The model is the same as in figure 1, except that the input (and output) of the model now contains multiple consecutive states $\mathbf{x}$, stacked along a single dimension.

latent space should be at least as big as the input space in order for the encoder to be invertible), it is more convenient to do so when the original dimension $n$ of the input space $\mathcal{X}$ is small. In this regard, the case of univariate time series (i.e. $n = 1$) is of particular interest. A graphical representation of an AIKAE model using this delay embedding strategy is shown on figure 3. This representation can obviously be generalized to predictions over multiple time steps (of the variable $\mathbf{y}$, each representing $m$ time steps for the original variable $\mathbf{x}$), as in figure 2.

In practice, using such a delay embedding of the state may increase the predictability when the knowledge of a single observation $\mathbf{x}_t$ is not sufficient to predict the subsequent states, i.e. in cases where $\mathbf{x}_t$ is not actually the state variable of a dynamical system. The use of delay embedding is commonplace in data-driven signal processing, as the well-known Takens theorem (Takens, 1981), guarantees an increased predictability of the system when the size of the delay embedding increases.

In particular, the use of delay embedding for DMD was proposed by Tu et al. (2014). Along with the subsequent works of e.g. Le Clainche & Vega (2017); Kamb et al. (2020); Yuan et al. (2021), they demonstrated the ability to model a higher number of Koopman modes, and an increased robustness to noise in the observed data. However, to the best of our knowledge, our work is the first to propose using a large delay embedding for a neural network-based implementation of the Koopman operator. Experiments involving this approach are presented in section 5.

## 4    AIKAE as a variational data assimilation prior

Variational data assimiliation consists in inferring the full state of a system over time, by leveraging a set of partial and noisy observations as well as some prior knowledge on the dynamical behavior of the system, often in the form of a dynamical model. Concretely, the assimilated state is obtained by minimizing a variational cost that comprises a term of fidelity to the observed data and a term of fidelity to the prior knowledge. This cost is minimized using some form of gradient descent algorithm. Traditionally, the prior knowledge comes in the form of a complex physical model, which can be differentiated using *adjoint* methods (see e.g. Bannister (2017)). A rich line of work has recently investigated the minimization of a variational cost using autodifferentiation frameworks, either by re-implementing physical models in such frameworks (see Gelbrecht et al. (2023) for a review) or by substituting this physical prior by a data-driven one (Nonnenmacher & Greenberg, 2021; Fablet et al., 2021). We refer the interested reader to Cheng et al. (2023) for a review of the methods combining machine learning and data assimilation.

Here, we show how to use a pre-trained AIKAE model as a prior for variational data assimilation, taking inspiration from the work of Frion et al. (2024). In a few words, this method consists in finding the initial latent state of the model that enables to most closely fit a set of observed states with associated timestamps.

Although it was originally introduced for regular (non-invertible) KAE models, it can be straightforwardly adapted to IKAE and AIKAE models, and thus we hereafter explicit the AIKAE case only.

Suppose that we have at disposal a trained AIKAE model, with its components $(\phi, \chi) = \Phi$ and $\mathbf{K}$. In addition, we observe a trajectory of data through a set of $T$ points $(\mathbf{x}_{t_0}, ..., \mathbf{x}_{t_T})$, with the associated time indexes $(t_0, ..., t_T) \in \mathbb{N}^{T+1}$, supposed to be arranged in increasing order with $t_0 = 0$ for convenience. Note that the timestamps could be chosen to be non-integers if we use the matrix logarithm of $\mathbf{K}$, as explained in Frion et al. (2024). In order to fit the observed datapoints, one can solve the following optimization problem:

$$\mathbf{z}_0^* = \min_{\mathbf{z}_0 \in \mathbb{R}^d} \sum_{i=0}^{T} ||\Phi^{-1}(\mathbf{K}^{t_i}\mathbf{z}_0) - \mathbf{x}_{t_i}||^2. \tag{14}$$

This method corresponds to a strong-constrained variational data assimilation scheme, where the chosen dynamical prior is the pre-trained AIKAE model. In practice, it can be solved using autodifferentiation, leveraging the fact that the prior is fully differentiable. Once the (approximated) solution $\mathbf{z}_0^*$ is found, one can query the predicted state at any time $t$ by simply computing

$$\hat{\mathbf{x}}_t = \Phi^{-1}(\mathbf{K}^t \mathbf{z}_0^*). \tag{15}$$

Then, depending on the time steps $t$ for which we are interested in the predictions, this framework may enable to solve denoising, interpolation, forecasting or all these tasks at once. In particular, the denoising capabilities of a well-trained model stem from the assumption that any trajectory produced by this model is physically consistent. Thus, the model is expected to be unable to exactly fit a set of noisy observations $(\mathbf{x}_{t_0}, ..., \mathbf{x}_{t_T})$ through equation 14, but to instead produce the physically consistent trajectory that best matches these points, which should remove the noise in the observations.

In order to adapt this method to a pre-trained IKAE model with components $\phi$ and $\mathbf{K}$, one would simply have to substitute $\phi$ to $\Phi$ in equations 14 and 15. Interestingly, since the latent space of an IKAE is in bijection with the state space, the exact equality of its trajectory to an observation at one given timestamp deterministically gives the remaining of the time series. To illustrate this remark, suppose that we constrain the equality of the predicted initial state to the initial observation in equation 14. Then, we can solve

$$\mathbf{z}_0^* = \min_{\mathbf{z}_0 \in \mathbb{R}^d} \sum_{i=0}^{T} ||\Phi^{-1}(\mathbf{K}^{t_i}\mathbf{z}_0) - \mathbf{x}_{t_i}||^2$$
$$\text{s.t.} \quad \Phi^{-1}(\mathbf{z}_0) = \mathbf{x}_0. \tag{16}$$

For an AIKAE, we have that the constraint $\Phi^{-1}(\mathbf{z}_0) = \mathbf{x}_0$ is respected if and only if $\mathbf{z}_0^i = \phi(\mathbf{x}_0)$. Thus, equation 16 is equivalent to an unconstrained optimization problem on $\mathbf{z}_0^a \in \mathbb{R}^p$. If $\mathbf{K}$ has a nonzero upper-right block (i.e. if different values of $\mathbf{z}_0^a$ can influence the invertible parts of the subsequent latent states in different ways), then multiple trajectories are admissible, making this problem nontrivial. In contrast, when adapting equation 16 for an IKAE, since there is no augmentation encoder, the only possible value for $\mathbf{z}_0$ is $\mathbf{z}_0^* = \phi(\mathbf{x}_0)$, which is the same one as in the direct inference in equation 8, taking no account of any of the subsequent observations. Overall, one can see that an AIKAE can produce several different trajectories that exactly match an observed initial state while an IKAE is not able to do so.

## 5 Long-term time series forecasting experiments

In this section, we present experiments on a set of popular long-term time series forecasting datasets. Sometimes called the "Informer benchmark" as a reference to the work of Zhou et al. (2021) that popularised it, it is comprised of the ETT datasets (including the subsets ETTh1, ETTh2, ETTm1, ETTm2), ECL, Exchange, Traffic and Weather. These datasets have been extensively used in the last few years to evaluate the performance of the recently introduced long-term time series forecasting models, including Transformers (Zhou et al., 2022; Nie et al., 2023), convolution-based methods (Wu et al., 2023) and linear models (Zeng et al., 2023). We refer the interested reader to Wang et al. (2024) for a recent assessment of the rapidly evolving state of the art on this benchmark. The long-term time series forecasting task consists in predicting

the state of a time series over a prediction length of $T_P$ timesteps, using as input a lookback window of $T_L$ preceding states. In practice, $T_L$ and $T_P$ are typically in the order of 100 time steps. Although the considered datasets consist in multivariate time series, it has been observed that using the information of a single variable over all time steps in a lookback window enables to obtain better performance than when considering the information of all variables at a single time step. For this reason, some of the best performing methods consist in either only one single univariate model that is used on every variable of the dataset (Zeng et al., 2023; Li et al., 2023), or in one univariate model for each variable (Nie et al., 2023). In particular, it has been repeatedly observed (see e.g. Zeng et al. (2023); Li et al. (2023); Toner & Darlow (2024); Han et al. (2024)) that simple linear models significantly outperform early Transformer models such as Informer (Zhou et al., 2021) on this benchmark. In addition, as underlined by Wang et al. (2024), sequential models such as long short-term memory networks (LSTM, Hochreiter (1997)) typically struggle to capture the long-term relationships compared to models that process the lookback window all at once. Using these insights, we propose to solve the long-term time series forecasting task with univariate delayed Koopman autoencoders, as described in subsection 3.2, rather than with a classical KAE that would use a single (multivariate) observation as its state space. Interestingly, a delayed KAE model may be seen as a generalization of the simplest linear model proposed by Zeng et al. (2023). In a few words, this linear model consists in directly finding a matrix $\mathbf{W} \in \mathbb{R}^{T_P \times T_L}$ representing a linear relationship between the observed lookback window $\mathbf{X} \in \mathbb{R}^{T_L}$ and the corresponding output $\mathbf{Y} \in \mathbb{R}^{T_P}$. Although $\mathbf{W}$ is found through stochastic gradient descent, this method is reminiscent of DMD with a delay embedding. Indeed, when supposing $T_L = T_P$, these two approaches are equivalent. In this regard, our delayed KAE approach is an additional generalization where the linear relationship is computed in a latent space defined by a nonlinear encoding through $\phi$ of the delay embedded state, rather than directly on this state. Thus, it will be of particular interest to assess whether the addition of a nonlinear encoder with an IKAE or AIKAE model enables to improve the forecasting performance, knowing that linear models have been observed to perform surprisingly well for the datasets that we consider.

We compare our delayed IKAE and AIKAE models against a set of strong and recent baselines representing several popular classes of models for long-term time series forecasting:

- The DLinear model (Zeng et al., 2023) is a variant of the previously discussed linear model, which leverages a trend-season decomposition of the lookback observations rather than the direct lookback window of observations.

- PatchTST (Nie et al., 2023) is a Transformer model, which decomposes the input time series into patches each containing information on several time indexes. It also treats each channel of the multivariate time series independently instead of mixing their information.

- Timesnet (Wu et al., 2023) is a convolution-based method. It consists in building 2-dimensional representations of the time series by reshaping the input data according to its main frequencies, and processing these representations using convolutional neural networks.

- iTransformer (Liu et al., 2024) is a Transformer model in which the feature and time dimensions are switched, which has been shown to enable better performance than all of the previously proposed variants of the Transformer model.

Following standard evaluation conditions (see e.g. Wu et al. (2023); Liu et al. (2024)), we test our IKAE and AIKAE models with a lookback window of size $T_L = 96$, and 4 lengths $T_P$ of the prediction window: 96, 192, 336, 720. Thus, the size of the invertible part of the latent space is $T_L = 96$. For AIKAE, the augmentation part of the latent space is of size 32, leading to a global latent space of size 128. We use reversible instance normalization (RevIN, Kim et al. (2021)) for IKAE and AIKAE, as it was reported to improve the performance of multiple long-term time series forecasting models. RevIN performs a channel-wise normalization of the input data, with 2 learnable parameters for each channel. In order to ensure the reproducibility of our results, we use a fixed random seed to initialise all IKAE and AIKAE models. The training is performed with the Adam algorithm with a learning rate of $10^{-3}$ and momentum parameters $\beta = (0.9, 0.999)$. A detailed account of the architectures and hyperparameter search for our models is deferred to appendix D.

Table 1: Forecasting mean squared errors (MSEs) and mean absolute errors (MAEs) for various models and long-term forecasting tasks. For each dataset, we use a lookback window of size $T_L = 96$ and prediction horizons $T_P$ of sizes $96, 192, 336, 720$. IKAE and AIKAE are our own implementations, while we use the results reported by Liu et al. (2024) for all other models. For each task and metric, the best result is in bold and the second best result is underlined.

| Model | | IKAE | | AIKAE | | iTransformer | | PatchTST | | TimesNet | | DLinear | |
|---|---|---|---|---|---|---|---|---|---|---|---|---|---|
| Metric | | MSE | MAE | MSE | MAE | MSE | MAE | MSE | MAE | MSE | MAE | MSE | MAE |
| ECL | 96 | 0.159 | 0.252 | 0.153 | 0.247 | **0.148** | **0.240** | 0.181 | 0.270 | 0.168 | 0.272 | 0.197 | 0.282 |
| | 192 | 0.173 | 0.265 | 0.167 | 0.259 | **0.162** | **0.253** | 0.188 | 0.274 | 0.184 | 0.289 | 0.196 | 0.285 |
| | 336 | 0.189 | 0.282 | 0.184 | 0.277 | **0.178** | **0.269** | 0.204 | 0.293 | 0.198 | 0.300 | 0.209 | 0.301 |
| | 720 | 0.227 | 0.313 | 0.223 | **0.310** | 0.225 | 0.317 | 0.246 | 0.324 | **0.220** | 0.320 | 0.245 | 0.333 |
| Traffic | 96 | 0.460 | 0.313 | 0.431 | 0.297 | **0.395** | **0.268** | 0.462 | 0.295 | 0.593 | 0.321 | 0.650 | 0.396 |
| | 192 | 0.476 | 0.317 | 0.449 | 0.302 | **0.417** | **0.276** | 0.466 | 0.296 | 0.617 | 0.336 | 0.598 | 0.370 |
| | 336 | 0.496 | 0.327 | 0.467 | 0.311 | **0.433** | **0.283** | 0.482 | 0.304 | 0.629 | 0.336 | 0.605 | 0.373 |
| | 720 | 0.524 | 0.343 | 0.499 | 0.329 | **0.467** | **0.302** | 0.514 | 0.322 | 0.640 | 0.350 | 0.645 | 0.394 |
| Weather | 96 | **0.157** | **0.209** | 0.160 | 0.211 | 0.174 | 0.214 | 0.177 | 0.218 | 0.172 | 0.220 | 0.196 | 0.255 |
| | 192 | **0.193** | **0.245** | 0.195 | 0.245 | 0.221 | 0.254 | 0.225 | 0.259 | 0.219 | 0.261 | 0.237 | 0.296 |
| | 336 | 0.237 | 0.281 | **0.237** | **0.281** | 0.278 | 0.296 | 0.278 | 0.297 | 0.280 | 0.306 | 0.283 | 0.335 |
| | 720 | 0.317 | **0.333** | **0.317** | 0.333 | 0.358 | 0.347 | 0.354 | 0.348 | 0.365 | 0.359 | 0.345 | 0.381 |
| ETTh1 | 96 | 0.386 | 0.399 | 0.386 | **0.398** | 0.386 | 0.405 | 0.414 | 0.419 | **0.384** | 0.402 | 0.386 | 0.400 |
| | 192 | 0.437 | 0.433 | **0.435** | 0.432 | 0.441 | 0.436 | 0.460 | 0.445 | 0.436 | **0.429** | 0.437 | 0.432 |
| | 336 | **0.480** | **0.457** | 0.482 | 0.459 | 0.487 | 0.458 | 0.501 | 0.466 | 0.491 | 0.469 | 0.481 | 0.459 |
| | 720 | 0.575 | 0.522 | 0.561 | 0.516 | 0.503 | 0.491 | **0.500** | **0.488** | 0.521 | 0.500 | 0.519 | 0.516 |
| ETTh2 | 96 | 0.301 | 0.348 | 0.300 | 0.346 | 0.297 | 0.349 | **0.288** | **0.338** | 0.340 | 0.374 | 0.333 | 0.387 |
| | 192 | **0.365** | 0.394 | 0.365 | **0.394** | 0.380 | 0.400 | 0.388 | 0.400 | 0.402 | 0.414 | 0.477 | 0.476 |
| | 336 | 0.394 | 0.422 | **0.387** | **0.416** | 0.428 | 0.432 | 0.426 | 0.433 | 0.452 | 0.452 | 0.594 | 0.541 |
| | 720 | **0.414** | **0.443** | 0.429 | 0.450 | 0.427 | 0.445 | 0.431 | 0.446 | 0.462 | 0.468 | 0.831 | 0.657 |
| ETTm1 | 96 | 0.328 | 0.364 | **0.322** | **0.360** | 0.334 | 0.368 | 0.329 | 0.367 | 0.338 | 0.375 | 0.345 | 0.372 |
| | 192 | 0.365 | 0.389 | **0.364** | 0.387 | 0.377 | 0.391 | 0.367 | **0.385** | 0.374 | 0.387 | 0.380 | 0.389 |
| | 336 | 0.390 | **0.405** | **0.390** | 0.406 | 0.426 | 0.420 | 0.399 | 0.410 | 0.410 | 0.411 | 0.413 | 0.413 |
| | 720 | 0.463 | 0.443 | 0.465 | 0.443 | 0.491 | 0.459 | **0.454** | **0.439** | 0.478 | 0.450 | 0.474 | 0.453 |
| ETTm2 | 96 | 0.184 | 0.266 | 0.185 | 0.267 | 0.180 | 0.264 | **0.175** | **0.259** | 0.187 | 0.267 | 0.193 | 0.292 |
| | 192 | 0.255 | 0.313 | 0.246 | 0.307 | 0.250 | 0.309 | **0.241** | **0.302** | 0.249 | 0.309 | 0.284 | 0.362 |
| | 336 | **0.297** | **0.343** | 0.303 | 0.344 | 0.311 | 0.348 | 0.305 | 0.343 | 0.321 | 0.351 | 0.369 | 0.427 |
| | 720 | **0.382** | **0.393** | 0.385 | 0.393 | 0.412 | 0.407 | 0.402 | 0.400 | 0.408 | 0.403 | 0.554 | 0.522 |

The full results are reported in table 1. The Exchange dataset is excluded from these results since it is the only dataset for which none of the tested models is able to clearly beat the persistence, which is a trivial method consisting in copying the last observed value from the lookback window to the whole prediction window. Classically, models that do not perform better than persistence are considered to contain no relevant information for the task at hand. Thus, we consider comparisons between methods that do not beat the persistence to be irrelevant. Extended results including the Exchange dataset and the persistence baseline can be found in appendix E.1.

From table 1, one can see that the AIKAE model clearly outperforms the IKAE model on the datasets for which the amount of training data is the highest, i.e. ECL and Traffic. In contrast, for the other datasets, AIKAE and IKAE obtain very similar results, which suggests that inflating the latent space with an augmentation encoding brings no improvement to the IKAE. Besides, both the IKAE and AIKAE models obtain very competitive performance against the other methods, ranking first or second for numerous tasks.

Our results seem promising considering the low numbers of parameters and overall simplicity of our methods. We emphasize that the number of parameters of our models depend on the lookback window length $T_L$, but not on the prediction length $T_P$, since longer predictions are simply obtained by autoregressive multiplications in the latent space. Thus, our models evaluated in table 1 have at most 110K parameters for IKAE and 175K parameters for AIKAE (for some tasks the parameter count is lower, due to the hyperparameter search described in appendix D). This is two orders of magnitude below most of the Transformer models, which have

tens of millions of parameters (see e.g. Zeng et al. (2023)) and comparable to linear models such as DLinear, which has around 140K parameters for $T_P = 720$. Besides, extensions of our models with classical time series processing tools such as the Fourier transform (Zhou et al., 2022) or seasonal-trend decomposition Zeng et al. (2023) might be interesting directions to study in order to further improve the results.

Additional results on this benchmark can be found in the appendices. Specifically, in appendix E.2, we study the influence of RevIN and of different encoder architectures, notably establishing the superiority of AIKAE to the IKAE with zero padding proposed in Meng et al. (2024). In appendix E.3, we study the performance of IKAE and AIKAE with varying lookback window sizes and show that, like linear models and in contrast to many Transformer models, their performance consistently improves with an increasing lookback window size. Our observation that the performance of older Transformer methods stagnates or even decreases as the lookback window size increases is consistent with the results of numerous previous works, e.g. Zeng et al. (2023); Nie et al. (2023); Liu et al. (2024). In appendix E.4, we study the sensitivity of the IKAE and AIKAE models to the random initialization of their parameters, and show that this sensitivity is moderate. In appendix E.5, we discuss the sensitivity of the performance with regards to some hyperparameters, namely the learning rate, the number of coupling layers of the invertible encoder and the width of these layers. Finally, in appendix E.6, we plot some predictions made by our models, along with the associated context windows and groundtruth.

## 6    Variational data assimilation on satellite image time series

We now move on to a training task involving long-term forecasting of satellite image time series at the pixel level. For reasons which will be shortly explained, this kind of data often involves missing observations, making it more challenging to handle than the time series data from section 5. Thus, contrarily to our approach for these previous experiments, we will not make use of the delay embedding strategy explained in section 3.2. Instead, we will train more classical Koopman autoencoder models, which we will then use as dynamical priors in the testing phase, using the assimilation approach described in section 4.

We work with a dataset of Sentinel-2 image time series, introduced by Frion et al. (2023b) and used as a variational data assimilation benchmark by Frion et al. (2024). These data differ from the time series datasets of the previous section in several ways. Most importantly, satellite images have multiple missing observations that are due to the presence of clouds between the observation satellite and the surface of the Earth. Since we are usually solely interested in modeling the surface, we find ourselves with the dilemma of either directly processing an irregularly sampled time series or interpolating the available observations as a pre-processing step. In the first case, the time series will be significantly more difficult to process. In particular, one cannot directly observe a lookback window of many previous observations in order to make a long-term prediction. In the second case, the time series is significantly easier to process, yet it is made partly synthetic by the interpolation pre-processing step, which will be learned by the model alongside the true distribution of the satellite data. Fortunately, as underlined by Frion et al. (2024), the Koopman autoencoder framework is more flexible than most time series processing methods thanks to its ability to learn an underlying continuous representation of the modeled system. However, in order to retain this flexibility, one cannot work with a large delay embedding as described in section 3.2, or use the other models from the benchmark of section 5. Instead, we will work with a more classical model, from which the input space is built out of only two consecutive observations, the second of which being used to compute a first order derivative. Indeed, as underlined in Frion et al. (2024), the access to a first order derivative enables to more easily compute short-term predictions since, when the evolution is smooth enough, one can already obtain a reasonably good approximation by using it to compute an explicit Euler scheme over one time step. Machine learning-based autoregressive forecasting models based on two previous observations are commonplace for tasks such as weather prediction: see e.g. Lam et al. (2023) and Oskarsson et al. (2024).

We now describe the forecasting benchmark that was introduced by Frion et al. (2024), on which we will test several variants of Koopman autoencoder models including our new AIKAE architecture. We have at disposal satellite images from two spatial areas: the forest of Fontainebleau and the forest of Orléans. The data from Fontainebleau, which is used as a training area, is regularly sampled in time thanks to a pre-processing Cressman interpolation step. To train a KAE model as a dynamical prior, we use $T_{train} = 242$

time steps of data, from an area of $150 \times 150$ pixels. The Sentinel-2 images that compose the dataset are multispectral, which means that they contain a richer spectral information than classical RGB images. Namely, the available information for each pixel and time step is a reflectance vector of size $L = 10$, including the classical red-green-blue spectral bands as well as 7 bands in the infrared domain. We emphasize that we work at the pixel level, which means that the input space of a model corresponds to the reflectance vector (and its first order derivative) of a single pixel, and that the pixel trajectories are assumed to all correspond to a same dynamical system.

After training, the trained model is used as a dynamical prior in a variational data assimilation framework, as discussed in section 4. The objective is to accurately predict the $T_{test} = 100$ steps of data that follow the window of training data, by leveraging the observations of this training window. This task is declined on the two areas discussed before. For the Fontainebleau training area, the time series is again regularly sampled in time, and only the capacity of the model to extrapolate to unseen time indexes is assessed. For the Orléans area, the data are not interpolated as a pre-processing step, and are thus irregularly sampled in time. In this case, all observations correspond to actual satellite measurements. The extrapolation task on this area tests not only the ability of the trained KAE model to extrapolate in time, but also to transfer its learned knowledge to a new spatial area with differing dynamics. Besides, the irregular sampling pattern of the observed data for this task is the precise reason for which one cannot resort to a model with delay embedding in this experiment.

It should be noted that directly training a model on irregularly-sampled data is preferable and that it is possible to do so with a KAE model, as demonstrated by Frion et al. (2024). However, we restrain our study to the case where training is performed on interpolated data in order to match the conditions of the main benchmark proposed by the authors. For this benchmark, we compare 4 variants of the Koopman autoencoder model:

- The base KAE model with 2 multilayer perceptron (MLP) networks as its encoder $\phi$ and decoder $\psi$, as described in Frion et al. (2024).

- An IKAE model leveraging a coupling layer normalizing flow model as its analytically invertible encoder, as proposed by Meng et al. (2024). More precisely, the encoder is implemented with the NICE architecture (Dinh et al., 2014). Note that substituting the NICE model by a non-volume preserving encoder such as RealNVP (Dinh et al., 2017) resulted in a less stable training procedure, constraining a reduction of the learning rate and leading to worse performance.

- An IKAE with zero padding, as suggested by Meng et al. (2024) (abbreviated IKAE-zp). This model is identical to the previous one except for the concatenation of zeros to the input state before entering the normalizing flow encoder, hence inflating the dimension of the latent space.

- Our AIKAE model, as described in subsection 3.1, where the latent dimension is inflated by the means of learning a second, non-invertible encoder $\chi$ (implemented as a MLP) rather than applying a fixed zero padding.

The size of the input space is $n = 2L = 20$, and the dimension is augmented by 16, leading to a latent space of size $d = 36$, for IKAE-zp and AIKAE. The latent dimension is set to $d = 32$ for KAE, and constrained to $d = n$ by design for IKAE. For each of these models, we train five instances corresponding to five parameter initializations with fixed random seeds. Following the recommendations of Frion et al. (2024), we design loss functions based on 4 terms: the prediction term, the reconstruction term, the linearity term and an additional orthogonality term. While the first 3 of these loss terms were proposed by Lusch et al. (2018) and are standardly used by multiple KAE implementations, the orthogonality term was proposed as a way to improve the long-term stability of the predictions, by ensuring that the norms of the latent states stay approximately constant through time. As previously mentioned, the 3 tested invertible models are trained without the reconstruction loss term since their reconstructions are exact by design. The training is performed with the Adam algorithm, with a learning rate of $10^{-3}$. We use weight decay with a coefficient of $10^{-6}$ for training the IKAE-zp and AIKAE models. For the other 2 models, we present results obtained with no weight decay since the usage of weight decay did not improve the performance.

Table 2: Mean squared errors (MSEs) and mean absolute errors (MAEs) obtained by averaging the performance of 5 instances of each model. The models are pre-trained on the Fontainebleau area, and then used as variational data assimilation priors, following equation 17. The KAE model was the only one to be subject to overfitting when assimilating the Orléans data, hence the additional row "Orléans (overfit)".

| Model | KAE | | IKAE | | IKAE-zp | | AIKAE | |
|---|---|---|---|---|---|---|---|---|
| Metric | MSE | MAE | MSE | MAE | MSE | MAE | MSE | MAE |
| Fontainebleau | 0.00112 | 0.0213 | 0.00128 | 0.0224 | **0.00106** | 0.0212 | 0.00108 | **0.0204** |
| Orléans (optimal) | 0.00346 | 0.0384 | 0.00382 | 0.0399 | 0.00324 | 0.0366 | **0.00297** | **0.0356** |
| Orléans (overfit) | 0.00403 | 0.0419 | N/A | N/A | N/A | N/A | N/A | N/A |

The testing procedure leverages the methods of section 4 by using the pre-trained model as a variational prior for data assimilation, with the motivation of producing a long-term forecast from an observed trajectory. Typically, for evaluating a trained AIKAE instance with components $\Phi$ and $\mathbf{K}$ on long-term forecasting the Fontainebleau data, we instantiate equation 14 as

$$\mathbf{z}_0^* = \min_{\mathbf{z}_0 \in \mathbb{R}^{d \times N \times N}} \sum_{t=0}^{T_{train}} ||\Phi^{-1}(\mathbf{K}^t \mathbf{z}_0) - \mathbf{x}_t||^2, \tag{17}$$

where $\mathbf{x}_t \in \mathbb{R}^{L \times N \times N}$ corresponds to the reflectance over an area of $N^2 = 100 \times 100$ pixels, which is included in the training data. Concretely, since the prior is a pixelwise model, this can be seen as $N \times N$ separate optimization problems. One could however perform a joint optimization with an additional spatial coherence prior, as proposed by Frion et al. (2024). After $\mathbf{z}_0^*$ is obtained, the forecasting mean squared error is computed as

$$\text{MSE} = \frac{1}{T_{test}} \sum_{t=T_{train}+1}^{T_{train}+T_{test}} ||\Phi^{-1}(\mathbf{K}^t \mathbf{z}_0^*) - \mathbf{x}_t||^2, \tag{18}$$

and the mean absolute error is computed in an analogous way. This procedure can be simply adapted to the other KAE variants by replacing $\Phi$ and $\Phi^{-1}$ by $\phi, \psi$ or $\phi^{-1}$ when necessary. The procedure is similar for the Orléans area, except that the assimilation cost and the metrics are computed only over time indexes where groundtruth observations are available.

Table 2 displays the obtained results. Additionally, in appendix F.3, we display and analyze forecasting results associated to this task for a randomly selected pixel. When extrapolating on the training Fontainebleau area, one can see that the performances of the models are relatively even, except for the IKAE model. Its worse performance may be attributed to the reduced size of its latent space, which limits its ability to find a proper linear representation of the system. The IKAE-zp variant seems to partially alleviate this issue as it obtains significantly better performance. The test Orléans area exhibits a stronger contrast between the tested models. On this area, one can see that the AIKAE model performs best, followed by the IKAE-zp.

Interestingly, the base KAE model was the only variant which was observed to overfit on its assimilated data on the Orléans area. In other words, the latent initial state $\mathbf{z}_0^*$ from equation 17 is not necessarily the best initial state for minimizing the forecasting error of equation 18. Concretely, as explained in section 4, we find an approximation to $\mathbf{z}_0^*$ by minimizing the associated variational cost using automatic differentiation with the Adam optimizer. We observe that the best extrapolation performance is obtained by using a relatively small learning rate and fewer gradient descent steps, resulting in a suboptimal minimization of the variational cost of equation 17. In contrast, for all other models, the best extrapolation result is always obtained by minimizing the variational cost, which means that the assimilation scheme does not overfit the assimilated data. To illustrate this difference, we report two results for the KAE model on the Orléans area: the first one (optimal) is with the optimal variational assimilation hyperparameters, as reported by Frion et al. (2024). The second one (overfit) is with the hyperparameters that minimize the cost of equation 17. The tendency to overfit on the assimilated data is a critical flaw in practice since, in real conditions, one cannot fit the assimilation hyperparameters using future data which are actually not known, and one will simply choose the hyperparameters that best fit the available data. Since the conditions for training and testing all 4 models

are very similar, the fact that the other models do not overfit the assimilation data should be attributed to an increased regularity enabled by their invertible encoders.

Additional results on AIKAE models for satellite image time series forecasting are presented in appendix F. More precisely, we study the influence of the size of the augmentation encoding $\mathbf{z}_t^a$ in section F.1, and variants of the linearity loss in section F.2. In a few words, the conclusions of these experiments are that 1) the performance of AIKAE models increases with the augmentation size until it eventually decreases due to overfitting, and 2) it is important to compute the linearity loss over both the invertible and the augmentation parts of the AIKAE encoding, with an unweighted least squares implementation.

## 7 Conclusion

We have experimentally shown that, for the recently introduced invertible Koopman autoencoder (IKAE) models, the analytical invertibility of the encoder can limit the practical ability to learn a Koopman invariant subspace, i.e. a latent space in which the system dynamics can be described linearly. Consequently, we proposed to augment these models with a new non-invertible encoder, resulting in our model: augmented invertible Koopman autoencoder (AIKAE). We have shown how recently proposed variational data assimilation schemes leveraging Koopman autoencoder models can be easily extended to the AIKAE architecture, enabling to work in difficult contexts where the observed data may be incomplete and noisy. Additionally, we proposed to design Koopman autoencoder models with a delay embedding, in order to solve long-term time series forecasting tasks in ideal settings where the data is regularly sampled with no missing information, and a large number of past states are observed. We showed that the AIKAE model performs equally or better than the IKAE, both in these ideal settings and in more difficult settings related to satellite image time series. Additionally, we showed that our AIKAE with delay embedding performs competitively with recent concurrent methods on a popular long-term time series forecasting benchmark.

A potential direction for future work would be to design stochastic Koopman autoencoder models, leveraging the likelihood computation abilities brought by the coupling-layer normalizing flows that compose the (augmented) invertible Koopman autoencoders. Besides, regarding our delay embedding strategy, it would be of interest to test alternative approaches where the input to the model is a condensed representation of several time steps (e.g. using mean pooling or convolution) rather than a direct stack of these steps, in order to reduce the computational cost as well as to increase the robustness to missing data.

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

# A  Reconstruction abilities of older Koopman-based methods

For Dynamic Mode Decomposition (DMD, Schmid (2010)), which is the most popular and well-established method for finding an approximation to the Koopman operator, the set of measurement functions is simply chosen to be the set of canonical measurement functions constituting the state variable. Hence, this method implicitly assumes that the dynamical system under study evolves linearly, or at least that accurate short-term predictions can be made by a linear model. In this case, the predictions are made directly in the state space, making its reconstruction unnecessary. A notable extension of DMD is the so-called extended Dynamic Mode Decomposition (Williams et al., 2015), which consists in manually designing a set of measurement functions that is likely to yield an approximate invariance by the Koopman operator. Common choices for these measurement functions are sets of polynomials of the state variables up to a chosen degree and sets of radial basis functions. The canonical measurement functions of the state are usually included in the hand-designed dictionary. This enables to trivially link the set of measurement functions to the state space by projecting on the appropriate variables. Although extended DMD is generally applied on low-dimensional dynamical systems, a high number of measurement functions is usually required to obtain accurate predictions, which is a limiting factor of this method in practice. Thus, some subsequent works (Li et al., 2017; Yeung et al., 2019) have proposed to replace the hand-designed dictionary of measurement functions by a lower-dimensional dictionary that is automatically learned by a neural network. In these models, the inferred Koopman invariant subspace is a concatenation of the fixed canonical measurement functions and of the ones that are learned by the neural network.

Although the inclusion of the state variables in the Koopman invariant subspace again enables to easily reconstruct the state vector after multiplication by $\mathbf{K}$, it may be detrimental to the actual linearity of the model. Indeed, depending on the dynamical system under study, there might not exist an (approximately) Koopman invariant subspace of low dimension that contains the state variables. This flaw has motivated the introduction of Koopman autoencoders Lusch et al. (2018), which do no constrain a direct inclusion of the input variables in their latent space.

# B  Stochastic modeling abilities of invertible Koopman autoencoders

Let us here outline some possible adaptations of invertible Koopman autoencoders for stochastic modeling, although we do not present associated experiments in the paper.

The determinant of the Jacobian matrix corresponding to a coupling-layer normalizing flow $\phi$ can be easily computed in practice, enabling to perform likelihood computations thanks to the well known change of variable formula:

$$p_X(\mathbf{x}) = p_Y(\phi(\mathbf{x})) \left| \det \frac{\partial \phi(\mathbf{x})}{\partial \mathbf{x}} \right|, \tag{19}$$

where $Y$ is defined in the latent space of the model.

Using these properties, coupling-layer normalizing flows were originally introduced as a generative model, enabling to link a complex probability distribution $p_X$, supposedly corresponding to a set of observed data, to a simple probability distribution $p_Y$, often chosen to be a standard Gaussian. Although the current existing works on invertible Koopman autoencoders only consider deterministic settings, these properties may be used to train invertible Koopman autoencoders in a stochastic context.

For example, one may estimate the probability distribution function of an advanced state $\mathbf{x}_{t+\tau}$ knowing that the state $\mathbf{x}_t$ is observed with an uncertainty corresponding to a Gaussian white noise with a covariance $\boldsymbol{\Sigma}_t \in \mathbb{R}^{n \times n}$. In this case, we would have that $\mathbf{x}_t$ is in fact a random variable, defined as

$$\mathbf{x}_t \sim \mathcal{N}(\boldsymbol{\mu}_t, \boldsymbol{\Sigma}_t). \tag{20}$$

Using equation 19, one can accordingly compute the probability density function of the associated encoding $\mathbf{z}_t = \phi(\mathbf{x}_t)$. From there, $\mathbf{z}_{t+\tau}$ is obtained as the multiplication of $\mathbf{z}_t$ by the square matrix $\mathbf{K}^\tau$, enabling for another easy change of variable since the Jacobian of this linear transformation is $\mathbf{K}^\tau$ itself. Finally, one can go back to the input space by simply reversing equation 19. As a conclusion, one may compute the

Table 3: Summary of the characteristics of the datasets of the Informer benchmark. The "Dataset length" entry contains the lengths of the training, validation and test subsets of data.

| Dataset | Channels | Sampling period | Dataset length | Information |
|---|---|---|---|---|
| ETTh1, ETTh2 | 7 | 1 hour | (8545, 2881, 2881) | Electricity |
| ETTm1, ETTm2 | 7 | 15 minutes | (34465, 11521, 11521) | Electricity |
| Weather | 21 | 10 minutes | (36792, 5271, 10540) | Weather |
| ECL | 321 | 1 hour | (18317, 2633, 5261) | Electricity |
| Traffic | 862 | 1 hour | (12185, 1757, 3509) | Transportation |
| Exchange | 8 | 1 day | (5120, 665, 1422) | Economy |

probability density function of $\mathbf{x}_{t+\tau}$ when $\mathbf{x}_t$ follows a known Gaussian distribution, which can enable to design a loss function that leverages the likelihood of subsequent states.

Conversely, following classical usage of normalizing flow models, one may consider the latent distribution to be Gaussian while the distribution in the input space is more complex. In this case, one may consider a variant of the IKAE where a variance encoder $\xi$ is trained along with the invertible normalizing flow encoder $\phi$, so that $\mathbf{z}_t$ is a random variable following:

$$\mathbf{z}_t \sim \mathcal{N}(\phi(\mathbf{x}_t), \xi(\mathbf{x}_t)), \tag{21}$$

where $\xi(\mathbf{x}_t) \in \mathbb{R}^n$ corresponds to the (positive) diagonal coefficients of a diagonal covariance matrix. In this case, $\mathbf{z}_{t+\tau} = \mathbf{K}^\tau \mathbf{z}_t$ is also Gaussian as a linear transformation of a Gaussian variable (one might consider adding Gaussian noise at each propagation step too), and one can again retrieve the probability density function of $\mathbf{x}_{t+\tau}$ with a final change of variable. Thus, one may construct a stochastic dynamical model of the considered system, trained with a likelihood criterion.

## C    Description of the long-term time series datasets

Here, we provide brief descriptions of the datasets constituting the benchmark from section 5. The ETT datasets (Zhou et al., 2021) are constituted from 7 factors of electricity transformers, including load and oil temperatures, recorded from July 2016 to July 2018. There are four subsets: ETTh1, ETTh2, ETTm1 and ETTm2. ETTh1 and ETTh2 have a sampling period of one hour, while ETTm1 and ETTm2 have a sampling period of 15 minutes. Weather (Wu et al., 2021) is constituted from 21 meteorological indicators, recorded every 10 minutes by the weather station of the Max Planck Biogeochemistry Institute in 2020. ECL (Wu et al., 2021) (also called "Electricity") is constituted from the hourly electricity consumptions of 321 clients, recorded from 2012 to 2014. Traffic (Wu et al., 2021) is composed of hourly occupancy rates of 863 roads in the San Francisco bay area, recorded by the California Department of Transportation from January 2015 to December 2016. Exchange (Wu et al., 2021) contains the daily exchange rates of eight different countries, ranging from 1990 to 2016.

In Table 3, we summarize the numbers of channels, sampling periods, time series lengths (including training, validation and test subsets), and nature of the information of these datasets. From this table, one can clearly see that the ECL and Traffic datasets contain much more channels than all of the other considered datasets. Concerning the loading and separation into train, validation and test splits for each dataset, we use the code from Zeng et al. (2023), corresponding to the same settings as for the benchmarks of Wu et al. (2023) and Liu et al. (2024) that the baseline methods use.

As explained in the main text, each of these datasets contains several simultaneously recorded channels of information, and thus one can theoretically combine the information of these channels in order to increase the predictive capabilities of a forecasting model. However, some recent methods (Zeng et al., 2023; Nie et al., 2023; Li et al., 2023) (along with the ones introduced in this paper) rely on a single univariate model, with performance challenging the strongest multivariate time series processing models.

# D   Implementation details for delayed Koopman autoencoders

Here, we describe in more detail the architectures of the delayed IKAE and delayed AIKAE models that produced the results on the first 2 columns of table 1. As explained in section 3.2, we make use of the delay embedding strategy, which means that the invertible encoding contains information on multiple time steps. Besides, we adopt the channel independence principle (Nie et al., 2023), so that an input state contains only information on one specific variable of the time series. Thus, since we use an input sequence length of 96 time steps for all experiments, the dimension of our input (and invertible encoding) is always $n = 96 \cdot 1 = 96$. The main component of both the IKAE and AIKAE is their invertible encoder $\phi$, which in these experiments is implemented as a NICE normalizing flow (Dinh et al., 2014). Using the terminology from Dinh et al. (2014), the NICE model is composed of $k$ coupling layers, each using the additive coupling law and a MLP network with one hidden layer of width $w$ and a leaky rectified linear unit nonlinearity (Maas et al., 2013). Following common modeling choices, the input to the MLP is composed of half of the input state variables, i.e. it is of size $n/2 = 48$. Thus, neglecting the bias parameters, the hidden layer of the MLP contains about $(n/2) \cdot w$ parameters and its output layer contains $w \cdot (n/2)$ parameters, for a total of approximately $wn$ parameters, and finally around $kwn$ trainable parameters when adding all the coupling layers of $\phi$. The decoder, being analytically obtained from this encoder, does not involve any additional trainable parameters. For the AIKAE models only, one additionally trains an augmentation encoder $\chi$, which is implemented as a MPL network. This MLP network comprises 3 linear layers, of width fixed to [256, 128, 32]. A ReLU nonlinearity Nair & Hinton (2010) is applied after the first 2 of these layers. Note that the width of the final layer is the size of the augmentation encoding, i.e. $p = 32$. This augmentation encoder contains approximately 60K trainable parameters. Finally, the Koopman matrix $\mathbf{K}$ adds $d^2$ parameters to the models, where $d = n = 96$ for the IKAE models and $d = n + p = 128$ for the AIKAE models.

A hyperparameter search is performed for each dataset and prediction length on the number $k$ of coupling layers (set in the range $k \in \{3, 4\}$) and the width $w$ of the hidden layer of the coupling functions (set in the range $w \in \{128, 256\}$). Thus, the number of parameters for the IKAE models varies approximately from 50K to 110K. Conversely, the number of trainable parameters for the AIKAE models varies approximately from 115K to 175K.

As mentioned in the main text, all models are trained with the Adam algorithm with a learning rate of $10^{-3}$ and parameters $\beta = (0.9, 0.999)$. As we observed a high sensitivity of the final performance on the batch size of the dataloader, for each task we search for the best batch size among the values $\{4, 128, 512\}$.

# E   Additional long-term time series forecasting results

## E.1   Extended forecasting results

Table 4 extends the results from table 1 by adding the persistence baseline and the exchange dataset. As mentioned in the main text, it shows that the tested models significantly outperform the persistence baseline on all datasets except for Exchange.

## E.2   Ablation study

In order to get insight on the performance of the delayed IKAE and AIKAE models, we now perform an ablation study. We focus on the influence of two components of our models: the nonlinear encoder and the usage of RevIN.

Concretely, we test eight different models. The IKAE and AIKAE models with RevIN correspond to the results reported in table 1. For each of these models, we train a variant where we do not use RevIN. In order to infer the interest of inflating the latent space with a second learned encoder rather than with zero padding as proposed by Meng et al. (2024), we also test IKAE models with zero padding, which are referred to as IKAE-zp, with or without RevIN. The size of the zero padding is 32, corresponding to the size of the augmentation encoding in AIKAE models. In addition, we test simple linear models, where the nonlinear encoder $\phi$ or $\Phi$ is simply replaced by an identity function, with (Li et al., 2023) or without (Zeng et al.,

Table 4: Forecasting mean squared errors (MSEs) and mean absolute errors (MAEs) for various models and long-term forecasting tasks. For each dataset, we use a lookback window of size $T_L = 96$ and prediction horizons $T_P$ of sizes $96, 192, 336, 720$. IKAE and AIKAE are our own implementations, while we use the results reported by Zeng et al. (2023) for the persistence baseline and by Liu et al. (2024) for all other models. For each task and metric, the best result is in bold and the second best result is underlined.

| Model | | IKAE | | AIKAE | | iTransformer | | PatchTST | | TimesNet | | DLinear | | Persistence | |
|---|---|---|---|---|---|---|---|---|---|---|---|---|---|---|---|
| Metric | | MSE | MAE | MSE | MAE | MSE | MAE | MSE | MAE | MSE | MAE | MSE | MAE | MSE | MAE |
| ECL | 96 | 0.159 | 0.252 | 0.153 | 0.247 | **0.148** | **0.240** | 0.181 | 0.270 | 0.168 | 0.272 | 0.197 | 0.282 | 1.588 | 0.946 |
| | 192 | 0.173 | 0.265 | 0.167 | 0.259 | **0.162** | **0.253** | 0.188 | 0.274 | 0.184 | 0.289 | 0.196 | 0.285 | 1.595 | 0.950 |
| | 336 | 0.189 | 0.282 | 0.184 | 0.277 | **0.178** | **0.269** | 0.204 | 0.293 | 0.198 | 0.300 | 0.209 | 0.301 | 1.617 | 0.961 |
| | 720 | 0.227 | 0.313 | 0.223 | **0.310** | 0.225 | 0.317 | 0.246 | 0.324 | **0.220** | 0.320 | 0.245 | 0.333 | 1.647 | 0.975 |
| Exch. | 96 | 0.081 | 0.202 | **0.080** | 0.201 | 0.086 | 0.206 | 0.088 | 0.205 | 0.107 | 0.234 | 0.088 | 0.218 | 0.081 | **0.196** |
| | 192 | **0.159** | 0.290 | 0.163 | 0.295 | 0.177 | 0.299 | 0.176 | 0.299 | 0.226 | 0.344 | 0.176 | 0.315 | 0.167 | **0.289** |
| | 336 | 0.331 | 0.435 | 0.320 | 0.429 | 0.331 | 0.417 | **0.301** | 0.397 | 0.367 | 0.448 | 0.313 | 0.427 | 0.305 | **0.396** |
| | 720 | 0.854 | 0.700 | 0.848 | 0.698 | 0.847 | 0.691 | 0.901 | 0.714 | 0.964 | 0.746 | 0.839 | 0.695 | **0.823** | **0.681** |
| Traffic | 96 | 0.460 | 0.313 | 0.431 | 0.297 | **0.395** | **0.268** | 0.462 | 0.295 | 0.593 | 0.321 | 0.650 | 0.396 | 2.723 | 1.079 |
| | 192 | 0.476 | 0.317 | 0.449 | 0.302 | **0.417** | **0.276** | 0.466 | 0.296 | 0.617 | 0.336 | 0.598 | 0.370 | 2.756 | 1.087 |
| | 336 | 0.496 | 0.327 | 0.467 | 0.311 | **0.433** | **0.283** | 0.482 | 0.304 | 0.629 | 0.336 | 0.605 | 0.373 | 2.791 | 1.095 |
| | 720 | 0.524 | 0.343 | 0.499 | 0.329 | **0.467** | **0.302** | 0.514 | 0.322 | 0.640 | 0.350 | 0.645 | 0.394 | 2.811 | 1.097 |
| Weather | 96 | **0.157** | **0.209** | 0.160 | 0.211 | 0.174 | 0.214 | 0.177 | 0.218 | 0.172 | 0.220 | 0.196 | 0.255 | 0.259 | 0.254 |
| | 192 | **0.193** | **0.245** | 0.195 | 0.245 | 0.221 | 0.254 | 0.225 | 0.259 | 0.219 | 0.261 | 0.237 | 0.296 | 0.309 | 0.292 |
| | 336 | 0.237 | 0.281 | **0.237** | **0.281** | 0.278 | 0.296 | 0.278 | 0.297 | 0.280 | 0.306 | 0.283 | 0.335 | 0.377 | 0.338 |
| | 720 | 0.317 | **0.333** | **0.317** | 0.333 | 0.358 | 0.347 | 0.354 | 0.348 | 0.365 | 0.359 | 0.345 | 0.381 | 0.465 | 0.394 |
| ETTh1 | 96 | 0.386 | 0.399 | 0.386 | **0.398** | 0.386 | 0.405 | 0.414 | 0.419 | **0.384** | 0.402 | 0.386 | 0.400 | 1.295 | 0.713 |
| | 192 | 0.437 | 0.433 | **0.435** | 0.432 | 0.441 | 0.436 | 0.460 | 0.445 | 0.436 | **0.429** | 0.437 | 0.432 | 1.325 | 0.733 |
| | 336 | **0.480** | **0.457** | 0.482 | 0.459 | 0.487 | 0.458 | 0.501 | 0.466 | 0.491 | 0.469 | 0.481 | 0.459 | 1.323 | 0.744 |
| | 720 | 0.575 | 0.522 | 0.561 | 0.516 | 0.503 | 0.491 | **0.500** | **0.488** | 0.521 | 0.500 | 0.519 | 0.516 | 1.339 | 0.756 |
| ETTh2 | 96 | 0.301 | 0.348 | 0.300 | 0.346 | 0.297 | 0.349 | **0.288** | **0.338** | 0.340 | 0.374 | 0.333 | 0.387 | 0.432 | 0.422 |
| | 192 | **0.365** | 0.394 | 0.365 | **0.394** | 0.380 | 0.400 | 0.388 | 0.400 | 0.402 | 0.414 | 0.477 | 0.476 | 0.534 | 0.473 |
| | 336 | 0.394 | 0.422 | **0.387** | **0.416** | 0.428 | 0.432 | 0.426 | 0.433 | 0.452 | 0.452 | 0.594 | 0.541 | 0.591 | 0.508 |
| | 720 | **0.414** | **0.443** | 0.429 | 0.450 | 0.427 | 0.445 | 0.431 | 0.446 | 0.462 | 0.468 | 0.831 | 0.657 | 0.588 | 0.517 |
| ETTm1 | 96 | 0.328 | 0.364 | **0.322** | **0.360** | 0.334 | 0.368 | 0.329 | 0.367 | 0.338 | 0.375 | 0.345 | 0.372 | 1.214 | 0.665 |
| | 192 | 0.365 | 0.389 | **0.364** | 0.387 | 0.377 | 0.391 | 0.367 | **0.385** | 0.374 | 0.387 | 0.380 | 0.389 | 1.261 | 0.690 |
| | 336 | 0.390 | **0.405** | **0.390** | 0.406 | 0.426 | 0.420 | 0.399 | 0.410 | 0.410 | 0.411 | 0.413 | 0.413 | 1.283 | 0.707 |
| | 720 | 0.463 | 0.443 | 0.465 | 0.443 | 0.491 | 0.459 | **0.454** | **0.439** | 0.478 | 0.450 | 0.474 | 0.453 | 1.319 | 0.729 |
| ETTm2 | 96 | 0.184 | 0.266 | 0.185 | 0.267 | 0.180 | 0.264 | **0.175** | **0.259** | 0.187 | 0.267 | 0.193 | 0.292 | 0.266 | 0.328 |
| | 192 | 0.255 | 0.313 | 0.246 | 0.307 | 0.250 | 0.309 | **0.241** | **0.302** | 0.249 | 0.309 | 0.284 | 0.362 | 0.340 | 0.371 |
| | 336 | **0.297** | **0.343** | 0.303 | 0.344 | 0.311 | 0.348 | 0.305 | 0.343 | 0.321 | 0.351 | 0.369 | 0.427 | 0.412 | 0.410 |
| | 720 | **0.382** | **0.393** | 0.385 | 0.393 | 0.412 | 0.407 | 0.402 | 0.400 | 0.408 | 0.403 | 0.554 | 0.522 | 0.521 | 0.465 |

Table 5: Forecasting mean squared errors (MSEs) and mean absolute errors (MAEs) of different models on three datasets, with lookback window $T_L = 96$ and forecasting horizon $T_P = 96$. For each dataset and metric, the best result is in bold and the second best result is underlined.

| Dataset | | Traffic | | Weather | | ETTm1 | |
|---|---|---|---|---|---|---|---|
| Metric | | MSE | MAE | MSE | MAE | MSE | MAE |
| AIKAE | with RevIN | **0.450** | 0.301 | 0.171 | **0.216** | **0.322** | **0.360** |
| | without RevIN | 0.497 | **0.299** | **0.167** | 0.226 | 0.350 | 0.386 |
| IKAE-zp | with RevIN | 0.452 | 0.304 | 0.174 | 0.219 | 0.331 | 0.365 |
| | without RevIN | 0.507 | 0.300 | 0.170 | 0.227 | 0.349 | 0.380 |
| IKAE | with RevIN | 0.460 | 0.313 | 0.174 | 0.220 | 0.328 | 0.364 |
| | without RevIN | 0.517 | 0.317 | 0.168 | 0.225 | 0.342 | 0.377 |
| Linear | with RevIN | 0.644 | 0.390 | 0.194 | 0.234 | 0.349 | 0.369 |
| | without RevIN | 0.649 | 0.397 | 0.201 | 0.266 | 0.345 | 0.377 |

2023) RevIN. We work with $T_L = T_P = 96$, which means that the linear model without RevIN may be seen as a dynamic mode decomposition (Schmid, 2010) with a delay embedding of size 96. We perform the ablation study on three datasets: Traffic, Weather and ETTm1. For each model, the architecture is kept the same for each dataset. The IKAE and AIKAE have an invertible encoder comprising 4 coupling layers of width 256. For the AIKAE, the augmentation encoder has the same architecture as in the main results. The optimizer is Adam with a learning rate of $10^{-3}$, and the batch size is 4 for the ETTm1 dataset and 32 for the two other datasets.

The results obtained by the eight described models are reported in table 5. Although we use our own implementation of the linear models in order to limit the risk of differing implementation choices influencing the study, we obtain consistent results with the implementations of Zeng et al. (2023) and Li et al. (2023). From these results, one can see that the addition of RevIN often (though not always) improves the performance of all backbone models. In addition, the gains obtained by using a more complex embedding appear to be complementary to the gains of RevIN. In particular, the delay AIKAE model without RevIN outperforms the delay IKAE without RevIN, which itself outperforms the linear model without RevIN. Thus, this study shows that resorting to an invertible nonlinear embedding of the input data improves the results compared to a simple linear model, and that the results are further improved when additionally increasing the dimension of this embedding with an AIKAE. Besides, the superiority of IKAE-zp to IKAE (either with or without RevIN) cannot be clearly established, and thus AIKAE remains the strongest of the tested KAE architectures in this benchmark.

### E.3 Influence of the lookback window size

It has been repeatedly observed in previous works (e.g. Zeng et al. (2023); Nie et al. (2023); Liu et al. (2024)) that many Transformer-based models for long-term time series forecasting do not benefit from an increased size $T_L$ of the lookback window. Indeed, for many of these models, the forecasting performance stagnates or even decreases as the length of the lookback window increases, which has been attributed to a distracted attention over the input. In contrast, simple linear models have been shown to greatly benefit from a longer window of observations. Thus, we now assess the performance of the IKAE and AIKAE models as the size of the lookback window increases. We work in the same setting as Zeng et al. (2023), where we evaluate the forecasting performance for a prediction window of size $T_P = 720$ according to varying input sizes $T_L$ from 48 to 720. For each lookback length, we train our two models as well as the DLinear model of Zeng et al. (2023) and 4 Transformer-based models: the base Transformer (Vaswani, 2017), Informer (Zhou et al., 2021), Autoformer (Wu et al., 2021) and FEDformer (Zhou et al., 2022). For the DLinear and Transformer models, we use the code of Zeng et al. (2023). For our IKAE and AIKAE models, in order to keep the experiment as fair as possible, we use the same architecture and training hyperparameters for every lookback length, i.e. 4 coupling layers of width 256 and a batch size of 512.

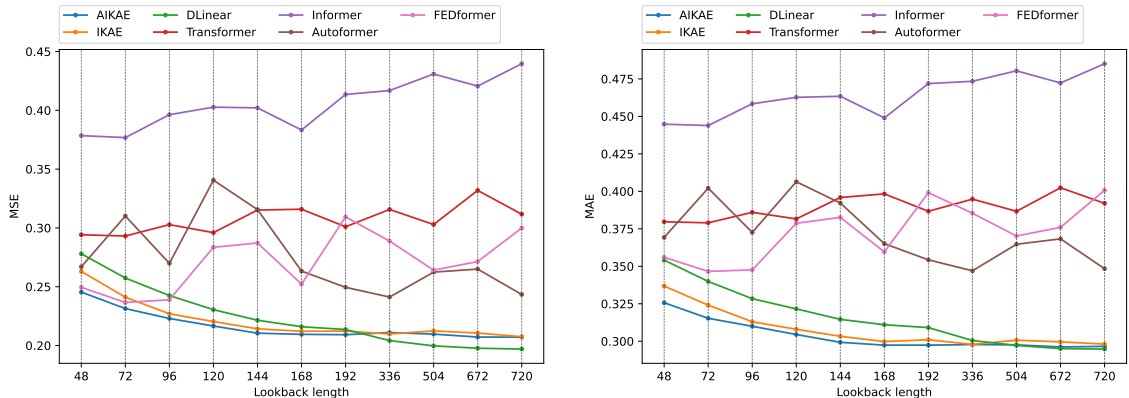

Figure 4: Mean squared error (left) and mean average errors (right) obtained by different models for predictions of $T_P = 720$ time steps on the ECL dataset, as a function of the lookback window size $T_L$.

Table 6: Mean and standard deviation of the MSE and MAE, over 5 random initializations of the parameters, for IKAE and AIKAE models on the ECL and Weather datasets.

| Model | | IKAE | | AIKAE | |
|---|---|---|---|---|---|
| Metric | | MSE | MAE | MSE | MAE |
| ECL | 96 | $0.1588 \pm 0.0005$ | $0.2518 \pm 0.0006$ | $0.1533 \pm 0.0010$ | $0.2469 \pm 0.0011$ |
| | 192 | $0.1727 \pm 0.0004$ | $0.2652 \pm 0.0003$ | $0.1667 \pm 0.0003$ | $0.2592 \pm 0.0003$ |
| | 336 | $0.1892 \pm 0.0004$ | $0.2818 \pm 0.0004$ | $0.1838 \pm 0.0006$ | $0.2768 \pm 0.0004$ |
| | 720 | $0.2267 \pm 0.0001$ | $0.3134 \pm 0.0006$ | $0.2235 \pm 0.0027$ | $0.3101 \pm 0.0020$ |
| Weather | 96 | $0.1573 \pm 0.0007$ | $0.2085 \pm 0.0008$ | $0.1603 \pm 0.0020$ | $0.2110 \pm 0.0009$ |
| | 192 | $0.1932 \pm 0.0018$ | $0.2450 \pm 0.0016$ | $0.1946 \pm 0.0028$ | $0.2451 \pm 0.0027$ |
| | 336 | $0.2374 \pm 0.0011$ | $0.2810 \pm 0.0010$ | $0.2374 \pm 0.0013$ | $0.2806 \pm 0.0011$ |
| | 720 | $0.3168 \pm 0.0014$ | $0.3328 \pm 0.0013$ | $0.3167 \pm 0.0013$ | $0.3335 \pm 0.0008$ |

The results of this experiment are summarized in figure 4. From this figure, one can see that none of the Transformer models is characterized by consistently decreasing error metrics as the size of the lookback window increases. Only the AIKAE, IKAE and DLinear models exhibit this behavior. While the AIKAE and IKAE models outperform the DLinear models for shorter lookback windows (as could be seen from the main results in table 1), DLinear performs best for longer lookback windows. Thus, this experiments shows that the delayed Koopman autoencoder models do not share the same flaws as many Transformer models, but still struggle to compete with linear models when a very large window of past observations is available.

### E.4 Sensitivity to the initialization

Here, we study the sensitivity of the delayed IKAE and AIKAE models to the initialization of their parameters before training. We focus on two datasets: ECL and Weather. For each prediction length of these datasets, we separately train 5 instances of IKAE and AIKAE models. These instances differ only by their initializations, each associated to a different fixed random seed. We report the mean and the standard deviations of the mean squared errors and mean absolute errors over these 5 seeds in table 6. From these results, one can see that the performance of our models are very stable across different initializations, since the standard deviations of the MSE and MAE are small in comparison to their associated means.

### E.5 Sensitivity to hyperparameter choices

Here, we study the performance of our models according to different training hyperparameters. Namely, we study the influence of the learning rate, the number of coupling layers in the invertible encoder $\phi$, and the hidden dimension of the MLP used in each of these coupling layers. We study learning rates of value

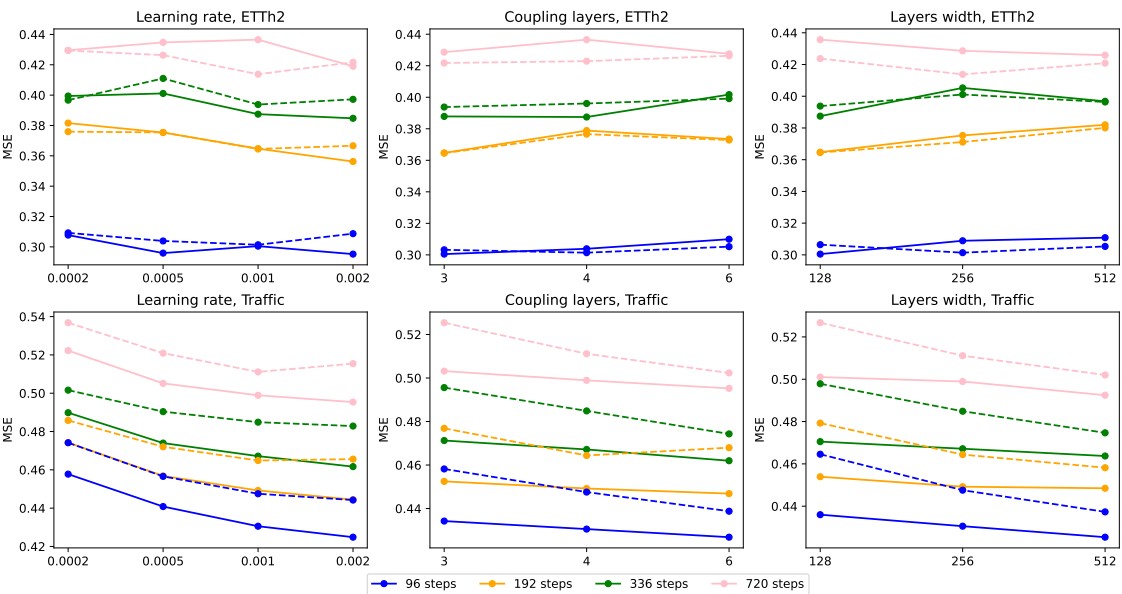

Figure 5: Summary of hyperparameter sensitivity analysis for delayed AIKAE and IKAE, on the ETTh2 (top) and Traffic (bottom) datasets. We study the influence of three hyperparameters: the learning rate (left), the number of coupling layers (middle) and the width of the layers (right). The full lines represent AIKAE models while the dashed lines represent IKAE models.

{0.0002, 0.0005, 0.001, 0.002} (as a reminder, all of our models in the main results are trained with learning rate 0.001). We consider numbers of layers in the range {3, 4, 6}, and hidden dimensions of values {128, 256, 512}. We focus our study on two datasets: ETTh2 and Traffic, for which we run our models in the same conditions as in the main results, with prediction lengths $T_P \in \{96, 192, 336, 720\}$. For the learning rate study, we always use the architecture which performed the best in the main results. Conversely, for the two other studies, the hyperparameters other than the one under study are fixed to their values in the best found configuration of the hyperparameter search detailed in section D.

The test MSE that we obtain are summarized in figure 5. These results show a relative robustness of the models to the choice of the learning rate. In fact, it seems that, although our main results are all obtained with a learning rate of 0.001, increasing this value to 0.002 (or including it in the hyperparameter search) might further improve the performance. Besides, one can see that increasing the number of parameters of the models (through the number or width of the layers) generally improves the performance on the Traffic dataset, but not necessarily on the ETTh2 dataset, which is consistent with the results of e.g. Liu et al. (2024). When specifically comparing AIKAE to IKAE models (respectively represented by full and dashed lines), one can see that the advantage of AIKAE is not clear for ETTh2 but significant for Traffic, which is consistent with the results of table 1. Interestingly, it appears that an AIKAE with layer width 128 performs slightly better than an IKAE with layer width 512 on the Traffic data. These models have respective parameter counts of around 125K and 210K, and thus one can say that the augmentation encoder enables a much greater parameter efficiency for this dataset.

### E.6 Case studies for long-term time series forecasting

In order for the reader to have a more intuitive idea of the quality of our model predictions, we now display some results obtained on randomly selected samples of the test subsets of datasets from the long-term time series forecasting benchmark. On figures 6 and 7, we respectively plot predictions on test samples from the ECL dataset and from the Weather dataset. In both cases, we show the predictions made over 96 time steps (left) and 336 time steps (right), with both an IKAE model and an AIKAE model, using the same context

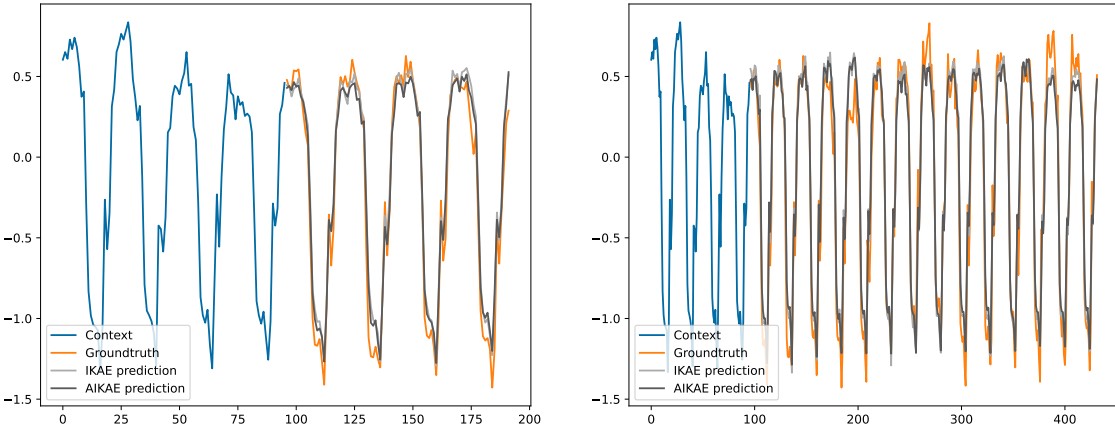

Figure 6: Predictions with AIKAE and IKAE models from a same context window in the test subset of the ECL dataset, over 96 time steps (left) and 336 time steps (right).

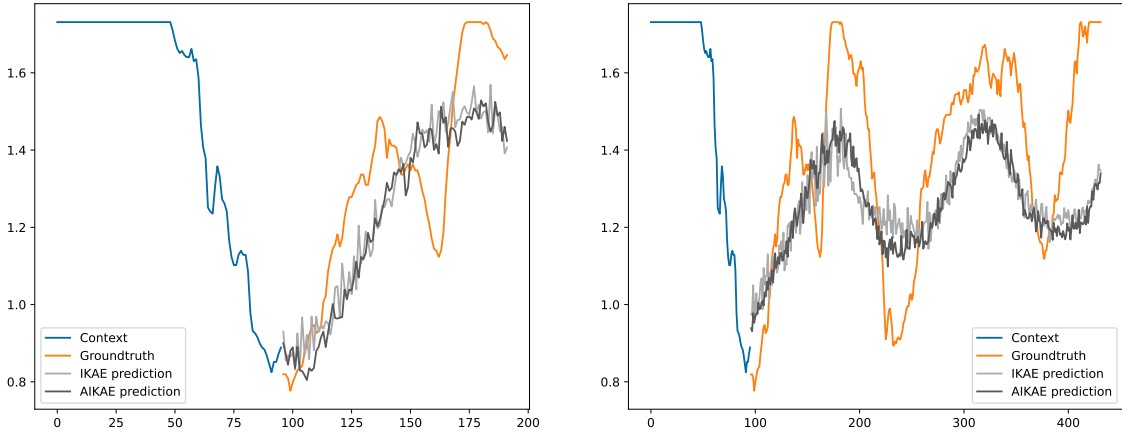

Figure 7: Predictions with AIKAE and IKAE models from a same context window in the test subset of the Weather dataset, over 96 time steps (left) and 336 time steps (right).

window of $T_L = 96$ time steps. From these figures, one can see that the predictions of the AIKAE and IKAE are qualitatively very similar.

# F Additional results for satellite image time series forecasting

## F.1 Influence of the size of the augmentation encoding

As stated in the main text, the worse results of the IKAE model compared to other Koopman autoencoder variants may be explained by the fact that the size of the input state is too low to obtain a good enough Koopman invariant subspace of the system. Here, we seek to obtain an empirical assessment of how the size of the augmentation encoding for AIKAE models affects the satellite image time series forecasting performance. As a reminder, the augmentation encoding for the AIKAE model results in table 2 is of size $p = 16$, resulting in a global latent space of size $d = n + p = 36$. Besides, the IKAE model may be seen as a special case of an AIKAE where $p = 0$. Building on these 2 cases, we train new AIKAE models with augmentation encoding sizes of 2, 4, 8 and 32. For all of these models, the other training hyperparameters are kept the same as in the AIKAE model from the main results. The reported results correspond to means over 5 random initializations of the parameters of the models. The full results are reported in table 7.

Table 7: Mean squared errors (MSEs) and mean absolute errors (MAEs) obtained by averaging the performance of 5 instances of each model. The models are pre-trained on the Fontainebleau area, and then used as variational data assimilation priors, following equation 17. We study varying sizes of the augmentation encoding in the AIKAE, reminding that an augmentation size of 0 corresponds to an IKAE models.

| Dataset | Fontainebleau | | Orléans | |
|---|---|---|---|---|
| Metric | MSE | MAE | MSE | MAE |
| Augmentation size 0 | 0.00128 | 0.0224 | 0.00382 | 0.0399 |
| Augmentation size 2 | 0.00110 | 0.0216 | 0.00362 | 0.0392 |
| Augmentation size 4 | 0.00109 | 0.0212 | 0.00359 | 0.0388 |
| Augmentation size 8 | **0.00105** | 0.0210 | 0.00314 | 0.0365 |
| Augmentation size 16 | 0.00108 | **0.0204** | **0.00297** | **0.0356** |
| Augmentation size 32 | 0.00118 | 0.0214 | 0.00365 | 0.0388 |

Several conclusions can be drawn from table 7. First, even a very low-dimensional augmentation encoding (i.e. augmentation size 2) can have a significant impact on the results. A potential explanation for this is that the augmentation variables of the latent space may represent more complex quantities since they are not structurally in bijection with the input. In addition, one can see that the performance gradually improves with increasing sizes of the augmentation encoding until the optimal value $p = 16$, and then degrades for the size $p = 32$. This shows that a large augmentation encoding might lead to an overfitting model. Overall, one can see that the performance of the model is very sensitive to $p$, which should therefore be chosen carefully in order to attain optimal results.

### F.2 Influence of the linearity loss on the augmentation encoding

In theory, in order for an AIKAE model to actually learn a Koopman invariant subspace of the studied dynamical system, one should have that the time propagation of an encoding at time $t$ by $\tau$ time steps leads to the encoding of the state at time $t + \tau$. This observation is the reason for the usage of a linearity loss function for Koopman autoencoder models, as explained e.g. by Lusch et al. (2018). With the notations of the AIKAE, the linearity loss, which we indeed use for training all Koopman autoencoder variants, can be written as

$$L_{lin}(\Phi, \mathbf{K}) = \mathbb{E}_{\mathbf{x}_t, \tau} ||\mathbf{K}^\tau \Phi(\mathbf{x}_t) - \Phi(\mathbf{x}_{t+\tau})||^2. \tag{22}$$

However, for the AIKAE in particular, we have noted in section 3.1 that only the invertible part of the encoding is directly related to the input space, and that the augmentation part of the encoding might be interpreted as containing the "static features" of the state. Thus, one might question the importance of actually ensuring that the augmentation part of the encoding follows a linear dynamics. Inspired by this observation, one may generalize equation 22 to a weighted version as

$$L_{lin,\alpha}(\Phi, \mathbf{K}) = \mathbb{E}_{\mathbf{x}_t, \tau} ||[\mathbf{K}^\tau \Phi(\mathbf{x}_t)]_{1:n} - \phi(\mathbf{x}_{t+\tau})||^2 + \alpha \, \mathbb{E}_{\mathbf{x}_t, \tau} ||[\mathbf{K}^\tau \Phi(\mathbf{x}_t)]_{n+1:d} - \chi(\mathbf{x}_{t+\tau})||^2. \tag{23}$$

In this variant, the case where $\alpha = 1$ corresponds to the unweighted version of equation 22, while $\alpha = 0$ means that only the invertible part of the encoding is expected to evolve linearly in time. In between these two cases, one might imagine choosing any value of $\alpha$ between 0 and 1 to determine the relative importance of the linearity of the invertible and augmentation parts of the encoding. A value $\alpha > 1$ would mean that it is more important for the augmentation part $\mathbf{z}_t^a$ than for the invertible part $\mathbf{z}_t^i$ to follow linear dynamics, which seems quite counter-intuitive.

In order to get some insight on these possible variants of the linearity loss function for the AIKAE model, we now examine three cases:

- the case $\alpha = 1$, as presented in the main results of table 2,

- the case $\alpha = 0$, where only the linearity of the invertible part $\mathbf{z}_t^i$ is required,

- the case $\alpha = 1/2$, where the linearity of $\mathbf{z}_t^i$ is twice as important as the linearity of $\mathbf{z}_t^a$.

Table 8: Mean squared errors (MSEs) and mean absolute errors (MAEs) obtained by averaging the performance of 5 instances of each model. The models are pre-trained on the Fontainebleau area, and then used as variational data assimilation priors, following equation 17. The models differ by the expression of the parameter $\alpha$ in their linearity loss term during training, following equation 23. The case $\alpha = 1$, corresponding to the main results of table 2, clearly leads to the best performance.

| Dataset | Fontainebleau | | Orléans | |
|---|---|---|---|---|
| Metric | MSE | MAE | MSE | MAE |
| $\alpha = 1$ | 0.00108 | 0.0204 | 0.00297 | 0.0356 |
| $\alpha = 1/2$ | 0.00114 | 0.0211 | 0.00335 | 0.0377 |
| $\alpha = 0$ | 0.00131 | 0.0223 | 0.00349 | 0.0382 |

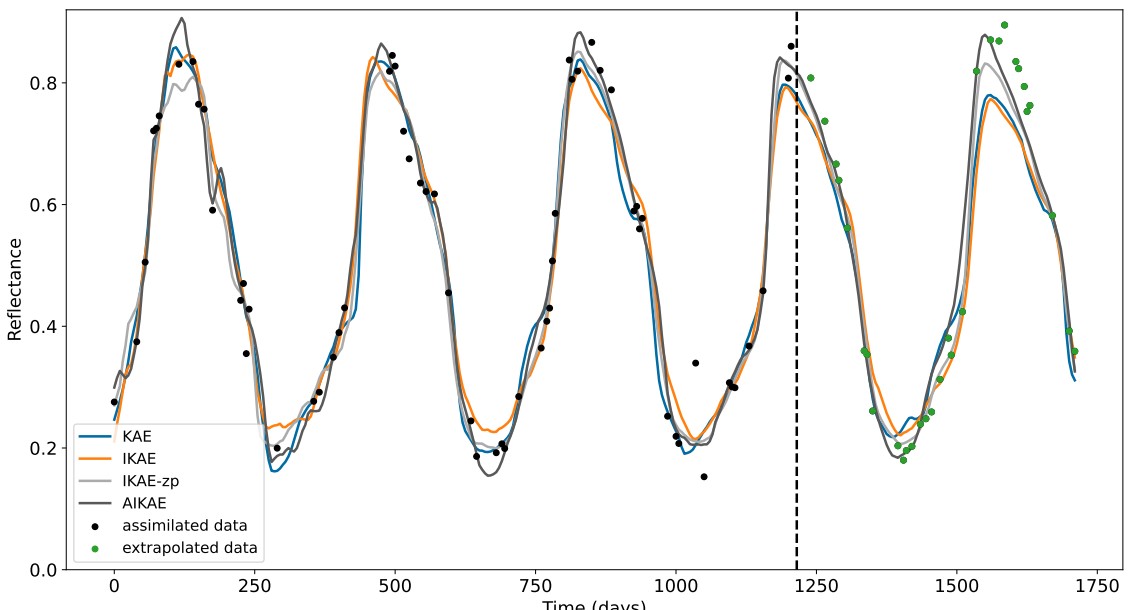

Figure 8: Assimilated trajectories by different models on the B7 spectral band, for a randomly selected pixel of the test Orléans area. The dashed vertical line marks the limit between the window of assimilated data (on the left) and the extrapolation window (on the right).

For each of these three cases, we train 5 instances of the AIKAE model starting from the exact same 5 random initializations, and otherwise following the same training procedure as in the main results. The means of the mean squared errors and mean absolute errors for each variant on the Fontainebleau and Orléans areas are listed in table 8.

From the results of table 8, one can see that the best performance are obtained with $\alpha = 1$, i.e. in the case where the linearity loss is implemented as an unweighted least squares calculation. The loss of accuracy when deviating from this case is significant. For the considered task, this result may be explained by the fact that the testing task is to perform a very long-term forecasting using an assimilation of the initial latent state while the training task involves naive forecasting on a relatively shorter timespan. In this context, the promotion of the linearity of the augmentation part of the encoding might be seen as a regularization of the model, bringing a significant boost to the test performance. One might expect that, in simpler setups, this observation might not always be true.

### F.3  Graphical results for satellite image time series forecasting

On figure 8, some assimilated trajectories are plotted on the B7 spectral band (i.e. the most energetic one for our data). The trajectories result from an assimilation of the data snapshots before the dashed line, using the best trained instance of respectively the KAE, IKAE, IKAE-zp and AIKAE models. On this example, the IKAE model clearly does not fit the assimilated data as well as the other models, which confirms our observation that the limited latent dimension might hurt the expressive power of this model. When it comes to extrapolating beyond the assimilated datapoints, the AIKAE model clearly performs best, followed by the IKAE-zp, which is consistent with the global results of table 2.

