# OpenReview forum: "Augmented Invertible Koopman Autoencoder for long-term time series forecasting"
_TMLR — Accepted by TMLR_

### Review · Reviewer_kcG2 · 2025-04-04

**Summary Of Contributions:**

This paper proposes a new Koopman-operator-based time-series processing model called Augmented Invertible Koopman Autoencoder (AIKAE). The model uses an invertible encoder based on coupling layer normalizing flows. It then augments this encoder with an additional trainable encoder and decodes only the channels that correspond to the coupling layer normalizing flows. In augmenting the encoder, the Koopman operator may be discretized in a much higher-dimensional space, which was a limitation of using only the coupling layer normalizing flows. Empirical results on the Informer Benchmark and a satellite image processing task show the promise of the model.

**Audience:**

Yes

**Claims And Evidence:**

Yes

**Requested Changes:**

1. On page 4, when you discussed the coupling-layer normalizing flows, the part that starts with "In addition, the determinant of the Jacobian matrix" and the next paragraph that starts with "Using these properties" sound a bit decoupled from the flow of the narratives. I think the paper can stand without them, and they do not serve very well as an introduction to readers unfamiliar with normalizing flows either. Please consider either compressing them or expanding them for better clarity.
2. For the model, my main question is as follows: As you mentioned earlier in the paper, the discretized Koopman matrix $\mathbf{K}$ does not exactly solve the equation in (4). I wonder, when you augment the state vector to have a dimension $d = n + p$, if there are differences between the errors on the first $n$ states and the last $p$ states. Intuitively, correctly matching the first $n$ states is more important than doing so for the last $p$ states because the first $n$ of them are what you used for decoding. In that sense, perhaps you can even explore a weighted least-square that favors matching some states over others. Maybe some discussions and empirical explorations along this direction can be insightful.
3. For the delayed autoencoder, you wrote "it is more convenient to do so when the original dimension $n$ of the input space $\mathcal{X}$ is small. In this regard, the case of univariate time series (i.e. $n = 1$) is of particular interest." Have you thought about other embedding ideas as opposed to simply stacking the previous inputs, such as using a mean pooling or a convolution?
4. In section 4, you mentioned that the model can be used for "denoising, interpolation, forecasting." It is straightforward to see how it can be used for interpolation and forecasting. Can you explain in the manuscript how denoising is possible? Does that relate to the implicit regularization of the model?
5. Also in section 4, you explained that the trajectory associated with the AIKAE model is nonunique due to the lack of injectivity from $\mathbf{z}$ to $\mathbf{x}$. In that case, is there a way to choose which trajectory to use? For example, does a different trajectory give you a different inductive bias of the model?
6. Minor issues:
   1. On page 3, please use $\cdot^+$ instead of $.^+$. Please use a double dash in the term "Moore--Penrose inverse."
   2. In (14), does this $d$ cause an overhead with the $d$ you used for the dimension of $\mathbf{z}$?

**Strengths And Weaknesses:**

Strengths:
   * The presentation of the paper is exceptionally good. I felt that the logic flows very smoothly, and the proposed model is well-motivated.
   * I personally found the proposed strategy very interesting. The idea of augmenting the encoder but only exploiting the invertible bit of the Koopman output is clever.

Weaknesses:
   * The paper has limited theoretical insights into the proposed model. In particular, I am interested in how the augmented state helps improve the approximation power of the Koopman unit $\mathbf{K}$ from an approximation theoretic point of view. I also have some questions regarding the new model listed below.
   * An ablation study would strengthen the paper, where you show how the performance depends on $p$, the output dimension of the augmented encoder $\chi$. I suspect that as $p$ gets large enough, the model will no longer benefit from increasing $p$.
   * The results on the Informer benchmark are promising but not extremely exciting.

Overall, I think this paper has good quality and look forward to a revised version.

---

> ### Author Response · Authors · 2025-05-06
> **Answer to reviewer kcG2 [part 1/2]**
>
> We are grateful to the reviewer for their thoughtful remarks and suggestions to improve our work. In particular, we are glad that they liked the presentation of our paper and found that our proposed model was well-motivated and interestering. Based on these comments, we have made numerous modifications to our manuscript, notably including significant improvements of the Informer benchmark results (thanks to a more extensive hyperparameter search, as suggested by reviewer gBJs) and new ablation studies for the satellite image time series forecasting experiments. We marked the modified/added text in red in our revised manuscript.
>
> Concerning the first expressed weakness on the limited theoretical insight into our model, we added a brief development on the linear algebra operations that characterize the influence of the augmentation part of the latent state on the invertible parts of subsequent latent states. This development is still limited, and we also consider that an approximation theory analysis would be very insightful. However, to the best of our knowledge, approximation theory results exist for extended dynamic mode decomposition and its kernel-based variants (e.g. [1], [2], [3]) but not for Koopman autoencoders. Thus, theoretical insights on neural network-based Koopman methods remain largely underexplored, and might be an interesting direction for future works.
>
> Let us now answer the requested changes one by one:
> 1. Acknowleding that this part did not fully fit in the general structure of the paper, we cut it in the main text and replaced it by a reference to a more detailed discussion in appendix B.
> 2. Inspired by this remark, we designed an ablation study for the satellite image times series experiment, in which the linearity loss term either ignores the augmentation part of the encoding or gives it only half as much importance as to the invertible part. This study finds that the model trained with full linearity loss achieves significantly better performance, but we carefully assume that the results might differ in other setups.
> 3. Other aggregation strategies such as mean pooling or strided convolutions might indeed be interesting strategies to exploit the information over a large time window without having to increase the latent dimension too much. In our case, we opted for directly using the stacked states as the input for several reasons: 1) this is relatively standard for dynamic mode decomposition and justified by the Takens theorem, as explained in section 3.2 from the text, and 2) we drew inspiration from methods such as DLinear which obtain good long-term forecasting results on the Informer benchmark with this strategy. Since the structure of the time series is preserved, this also enables to predict the evolution multiple time steps ahead with only one forecasting step in the latent space. However, using a more condensed representation of past observations (along with  the state at time t) might indeed be more appropriate in some cases, being more robust to missing data and less computationally demanding for multivarite time series. We mentioned in the conclusion that this would be an interesting direction for future works.
> 4. Denoising is possible by solving exactly the same optimization problem as for interpolation. It does indeed rely on the implicit regularisation brought by the model (not the AIKAE in particular, but any KAE). Basically, we consider that the model is simple enough for it not to be able to fit any set of observations. In fact, we expect that any trajectory produced by the model is physically consistent with the modeled dynamical system. Thus, when looking for the initial latent condition that best matches a set of noisy observations, we expect that the resulting trajectory will fit the observations as best as it can, but not at the expense of the physical constistency, which should have a denoising effect. We added a brief explanation for this in the text.
> 5. Yes, one may indeed choose which trajectory to use from a constrained initial state, for example by trying to get the best fit of subsequent observations, which is exactly the spirit of our equation 17 from the initial submission (equation 16 in revised version). In order to favor a desired inductive bias of the obtained trajectory, one may accordingly define a differentiable criterion to minimize in a similar optimization problem. It might also be possible to design an AIKAE model such that $\mathbf{z}_t^a$ contains some physical information on the trajectory on its own.

---

> > ### Author Response · Authors · 2025-05-06
> > **Answer to reviewer kcG2 [2/2]**
> >
> > 6.
> >     1. Thanks for pointing out these typographical mistakes, they are now corrected.
> >     2. Indeed, the choice of using the letter "d" here is not the best, since the quantity it refers to is actually the size of the input to the delayed model, while the model's latent dimension could indeed be the same in the case of an IKAE, but also larger in the case of an AIKAE. In order to improve the notation's clarity, we introduce a new notation instead, "$n'$", thus making an implicit connection to our notation $n$ for the original input dimension.
> >
> > [1] Korda, Milan, and Igor Mezić. "On convergence of extended dynamic mode decomposition to the Koopman operator." Journal of Nonlinear Science 28 (2018): 687-710.
> >
> > [2] Kostic, Vladimir, et al. "Learning dynamical systems via Koopman operator regression in reproducing kernel Hilbert spaces." Advances in Neural Information Processing Systems 35 (2022): 4017-4031.
> >
> > [3] Kostic, Vladimir, et al. "Sharp spectral rates for Koopman operator learning." Advances in Neural Information Processing Systems 36 (2023): 32328-32339.

---

### Review · Reviewer_gBJs · 2025-04-09

**Summary Of Contributions:**

This paper introduces the Augmented Invertible Koopman AutoEncoder (AIKAE), addressing the dimension limitation of existing invertible Koopman autoencoders (IKAEs) by augmenting the latent space with a non-invertible encoder. The AIKAE retains exact reconstruction via its invertible component while expanding the Koopman operator’s expressivity. Experiments on long-term forecasting benchmarks (e.g., ETT, Traffic, Weather) demonstrate AIKAE’s competitiveness with state-of-the-art methods like iTransformer and DLinear, particularly showing improved performance over IKAEs. Additionally, AIKAE excels in variational data assimilation for satellite image time series with missing observations, outperforming other Koopman variants. The model’s delayed embedding approach leverages large lookback windows, aligning with linear models’ scalability advantages. Code availability and theoretical grounding in Koopman operator theory further strengthen its contribution. However, gains over baselines are often marginal, and computational costs or hyperparameter sensitivity remain underexplored.

**Audience:**

Yes

**Claims And Evidence:**

Yes

**Requested Changes:**

(1)	As for weakness point 2, It’s necessary to clarify why list the "Variational data assimilation on satellite image time series" part.

(2)	As for weakness point 3, hyperparameters tuning or other techniques may help.

(3)	As for weakness point 4 and 5, it will be better if case study, hyperparameters analysis and complexity analysis provided.

**Strengths And Weaknesses:**

Strengths

(1)	This work combines invertible and non-invertible encoders to expand latent space without sacrificing reconstruction accuracy, and grounded in Koopman operator theory, addressing invariance and linear dynamics approximation.

(2)	It validated on both regularly sampled time series and irregularly sampled satellite imagery. And it matches or surpasses recent baselines (e.g., Transformers, DLinear) in long-term forecasting tasks.

(3)	This work has publicly released code that enhances reproducibility and community adoption.

Weaknesses:

(1)	The description in the methodology section is not sufficiently clear. Figure 1 only illustrates the transition from Xt to Xt+1, but fails to show feature extraction and fusion across different time steps. Additionally, the overall mapping from multiple input steps through feature extraction to multiple output steps is not clearly depicted. Details regarding each sub-module used in the method are also missing, making it challenging to grasp the methodology clearly at either the global or detailed level.

(2)	The paper's topic of long-term time series forecasting, but the motivation behind introducing the "Variational data assimilation on satellite image time series" part is insufficiently articulated. Satellite image time series appear inadequately connected to the general theme of long-term time series forecasting (such as topics mentioned in related work).

(3)	In the experimental results compared with baselines, too few indicators achieve comparable state-of-the-art performance.

(4)	No case study is provided to verify the prediction performance of the model in practical scenarios. Also, there is limited discussion on the sensitivity of the model's performance to hyperparameter choices, which is important for practical implementation.

(5)	No discussion of training/inference costs or scalability comparison with the listed baselines.

(6)	The potential for stochastic Koopman modeling is mentioned but not demonstrated.

---

> ### Author Response · Authors · 2025-05-06
> **Answer to reviewer gBJs**
>
> We thank the reviewer for their useful feedback on our manuscript. We appreciate that they underlined the interest of validating our methods on both regularly and irregularly sampled time series, and the public release of the code associated to our work. Based on the reviewer's feedback, we have made significant changes to the manuscript, which are marked in red in its new version. These changes include improvements of the main results and new experiments, which we believe represent significant improvements to the manuscript. We now move on to an individual answer to each of the reviewer's concerns.
>
> (1) We acknowledge that, although the details are explained in the text, having graphical representations of our long-term forecasting strategies might make the manuscript easier to understand. For this reason, we added a figure that graphically represents predictions over 2 time steps (figure 2), and a figure that graphically represents the delay AIKAE (figure 3). Concerning the detailed description of the submodules $\phi$ and $\chi$, we emphasize that the AIKAE model as introduced in section 3 is meant to allow for a great freedom of implementation of these submodules (only requiring $\phi$ to have an analytical inverse), and therefore a detailed description of a particular architecture choice does not belong here. Instead, we added a new appendix section (D) providing additional implementation details concerning the models used in our experiments. We also suggest looking into our available code if some points remain unclear.
>
> (2) We emphasize that the variational data assimilation is also used as a method to solve a long-term time series forecasting task, but in a context with missing data. From our point of view, the classical forecasting datasets such as ETT and Traffic correspond to a rather restrictive view of forecasting, where the data is regularly sampled with no missing data, low measurement noise, a high enough sampling frequency that aligns with structural cycles in the dynamics... In order to work on irregularly-sampled data in particular, one may simply remove some observations from these datasets with a chosen masking pattern, but we find it preferable to directly work on data for which some observations cannot be acquired, in which case the masking pattern is determined by the measurement constraints rather than arbitrarily: this is exactly the case for the Sentinel-2 datasets that we consider. We work on this benchmark as well as on a more classical long-term forecasting benchmark in the same manuscript, with a shared codebase, as an effort to encourage cross-connections between different fields of research.
> Regarding the related requested change (1), we added an introductory paragraph at the beginning of section 6 that sums up why the task differs from section 5 and how we adapt our methods accordingly.
>
> (3) Taking the advice of the reviewer, we obtained significant improvements of our performance thanks to an extensive hyperparameter search, which is detailed in appendix D. Our delayed IKAE and AIKAE models notably outperform all baselines by a considerable margin on Weather, and rank best in many other settings. Our github repository will shortly be updated accordingly.
>
> (4) One could argue that the satellite image time series experiment can be seen as a case study for a practical scenario. Regardless, we have added some graphical examples of long-term time series forecasting in the appendix section E.6. We have to admit that we are not sure to understand what is meant by "case study", so we are ready to address remaining concerns if a more specific expectation is expressed.
> We have added experiments on the sensitivity of our models to their parameter initialization and hyperparameter choices in the appendix sections E.4 and E.5.
>
> (5) There was a short discussion on the number of parameters and scalability of our models in section 5 of our submitted manuscript. In a few words, the order of magnitude of the parameter counts for our methods is the same as for linear models (which is 2 orders of magnitudes below Transformer variants such as Informer, Autoformer and FEDformer) and, in contrast to Transformer and linear methods, the complexity does not scale with the output window size. Unfortunately, among the numerous other requested changes, we did not find time to provide an extensive model efficiency comparison with recent methods.
>
> (6) Taking into account this remark as well as a similar one by reviewer kcG2, we decided to cut the development on stochastic models from the main text and refer to a more detailed discussion in appendix B instead.

---

> > ### Comment · Reviewer_gBJs · 2025-05-10
> > **Revision**
> >
> > Thank you for the comprehensive supplementary materials on methodological details, explanations of datasets, as well as the additional experiments on finetuning and hyperparameter sensitivity. I have no further comments.

---

### Review · Reviewer_FaCp · 2025-04-28

**Summary Of Contributions:**

The authors introduce Augmented Invertible Koopman AutoEncoder (AIKAE) which is an extension of Dynamic Mode Decomposition (DMD) technique by incorporating (1) invertible autoencoder and (2) further augmentation via additional observables. Both the autoencoder and observable mappings are certain types of deep learning models. By introducing both an invertible autoencoder and augementation, the authors show that AIKAE achieve roughly the state-of-the-art results.

**Audience:**

Yes

**Broader Impact Concerns:**

None.

**Claims And Evidence:**

Yes

**Requested Changes:**

1. The authors should verify the numerical precision of the inversion as in Weaknesss 2, either theoretically or numerically, or adjust the statement.

2. The authors should revisit the linear algebraic properties of the augmented DMD procedures, as observed in the Weakness 3. I recommend the authors analyze this further by analyzing the linear algebraic properties of the DMD augmentation.

3. I recommend the authors provide full details on how the compared models (transforms, convnets, etc) were trained and fine-tuned. If these were available in the literature, the authors should provide a broad summary. I believe the numerical results would be much more convincing if these details show that these models cannot be fine-tuned to yield better accuracy (or at least it is difficult to do so).

4. Page 2: "However, most of the time the Koopman invariance has to be approximated to a certain degree" is a bit unclear.

5. Page 4: "IKAE architecture had the inconvenient" : "inconvenience"?

6. Page 7: I recommend the authors provide a brief description of what the time-series dataset is. It is omitted in the current manuscript, I believe.

**Strengths And Weaknesses:**

Strengths

- The paper is very clearly written and this reviewer appreciates the nice exposition of the DMD framework and its extension.
- The introduced ideas are well-motivated, and the introduced method is explained clearly without burdening the readers with too much technicalities.

Weaknesses

1. I felt the introduction of augmented observables could be motivated better. If the invertible mapping is already transforming all the necessary information into latent variables, why is the augmentation necessary? The authors provide base this on learning behavior depending on the latent variable dimension in order to increase the redundacy in the data. However, if the mapping is invertible, such augmented information must be available in the latent variables $\mathbf{z}^i_t$ already. Further explanation would be helpful.

2. There are some mathematical observations that are incorrect, at least according to my first reading. The authors seem to be under the impression that because the autoencoder is invertible algebraically, the inversion can be done to machine precision (page 4: "enabling an exact reconstruction of an encoded state $\mathbf{x}$ up to machine precision). This is not true. When floating point arithmetic is used, even for exact algebraic expressions, conditioning affects the accuracy of the inverted result. For the authors' claim to be true, it should be back up by guarantees in the conditioning of the inverse.

3. The authors claim that, when the DMD matrix $\mathbf{K}$ learned from the latent data $[ \mathbf{z}^i_t, \mathbf{z}^a_t]$, "the augmented variables $\mathbf{z}^a_t$ influences the subsequent invertible parts of the encoding through the multiplication by $\mathbf{K}$ as long as the upper-right block of $\mathbf{K}$ is nonzero." This is a sufficient condition but not a necessary one; the upper-right block can be non-zero but $\mathbf{z}^a_t$ could lie in the nullspace. Therefore, having upper-right block of $\mathbf{K}$ non-zero does not imply that the augmented variables are being used at all. Furthermore, it is not difficult (mathematically speaking) to construct simple examples where the influence is non-zero, but is trivial.

4. The computational results can benefit from more details. The authors seems to be showing that, somehow the large transformer models underperform compared to the AIKAE (or IKAE) and that errors do not necessarily decrease when the "lookback length" is increased. This phenomena is a bit confusing. In general, longer time-series data used for prediction leads to better prediction results. These models appear to have more parameters, more room for hyperparameter tuning, etc, and it is not clear a prior whether these results (e.g. Figure 2) are exhibiting real bottlenecks in the learning procedure, or whether they have just not been fine-tuned enough.

---

> ### Author Response · Authors · 2025-05-06
> **Answer to reviewer FaCp [1/2]**
>
> We thank the reviewer for their insightful review, which helped us to improve the manuscript. We are pleased that they found our manuscript clearly written and our ideas well motivated.
>
> We hereafter provide answers to each of the expressed weaknesses:
> 1. The reason for the introduction of augmented variables is that the invertible encoding constrains the latent dimension to be the same as in the input, which might be insufficient to identify a Koopman invariant subspace (i.e. linear latent dynamics). There are concrete exemples in the literature (e.g. [8] equation 21, as mentioned in the manuscript), where adding just one redundant variable to the set of state variables (in this case the square of one of them) enables to obtain exact Koopman invariance. The last paragraph of section 2 and the first of section 3 address this motivation. However, acknowledging the fact that this motivation might not have been highlighted enough in the general presentation of the paper, we decided to add a mention of it in the conclusion. In addition, the new Appendix F.1, studying the performance of AIKAE as a function of the augmentation size, may provide further experimental insight in this regard.
> 2. Our formulation was indeed misleading. The state cannot necessarily be reconstructed up to machine precision, but it is still reconstructed with an approximation error that is solely due to numerical approximations, since the algebraic expression is an exact equality (which is usually not the case for neural autoencoders). As suggested, we adjusted our statement accordingly, by simply saying that the reconstruction is algebraically exact. The numerical reconstruction error is usually not discussed in the normalizing flow literature, and seems of little interest to us, which is why we do not think it deserves a detailed discussion.
> 3. The reviewer is right to point out that having a non-zero upper-right block of $\mathbf{K}$ is a necessary but insufficient condition for the augmentation part $\mathbf{z}^a_t$ of the encoding to influence the invertible part $\mathbf{z}^i_{t+1}$ and subsequent ones, since this augmentation vector might simply lie in the nullspace of the block. In fact, whatever the matrix $\mathbf{K}$, an augmentation vector of zero (i.e. in the nullspace of any matrix) will have no influence on subsequent invertible encodings. Yet, what we really meant to say is that different values of $\mathbf{z}^a_t$ may have differing impact on $\mathbf{z}_{t+1}^i$, as long as $\mathbf{K}$ has a non-zero upper-right block. We reformulated the text in order to make this more clear, with a short discussion about the nullspace of the upper-right block.
> 4. Our observation that many Transformer models perform sub-optimally on long-term time series forecasting are far from new. There is indeed a rich line of work (e.g. [1, 2, 3, 4]) demonstrating that simple linear models outperform many Transformer models on long-term time series forecasting, and discussing potential explanations for these conterintuitive results. In a few words, the Transformers might perform sub-optimally due to overfitting and shifting distributions between the train, validation and test subsets, to which simpler models are less sensitive.
> In particular, concerning the experiment with varying lookback lengths, we follow a very common setup of recent influential papers in the field. For example,  a series of similar results to ours can be found in figure 4 from the DLinear paper [1], figure 2 from  the PatchTST paper [5] and figure 6 from the iTransformer paper [6]. All of these figures show, like ours, stagnating or decreasing performance of older Transformer methods as the lookback size increases. It is obviously true that the theoretical capabilities of a model increase as more information is available, and yet it is a widely accepted belief that basic Transformer models do not benefit from it on this benchmark. This belief might be partly due to insufficient hyperparameter tuning (we re-used the code of [1]), and yet our models (and the ones from [1]) use the same hyperparameter settings for all lookback sizes, so at the very least one can say that Transformer models require far more tuning than simpler methods to reach their optimal performance.
> We added some additional references to [1,2,3,4,5,6] in the text and appendix in order to make it more clear that these claims are not new.

---

> > ### Author Response · Authors · 2025-05-06
> > **Answer to reviewer FaCp [2/2]**
> >
> > Finally, concerning the requested changes, we refer to the updated PDF for a detailed examination of our modifications, which are marked in red. We briefly comment on these below:
> > 1. We have adjusted our statement.
> > 2. We have made our statement more clear, by mentioning the nullspace of the upper-right block of K.
> > 3. As mentioned in the text, our baseline results are directly taken from table 10 of [6]. The results from [6] are themselves in part obtained from [7]. Both papers mention that they used the best reported hyperparameters from all of the reported baselines. For the sake of coherence and comparability, we consider that it is preferable to simply consider these results as baseline than to design a whole new setup. The interested reader can always refer to [6], [7] and the refereinces therein for details on the implementation of the baseline methods.
> > 4. We prolonged the sentence to make it more explicit.
> > 5. Corrected, thanks for pointing it out.
> > 6. We added appendix C, which gives a brief description of the datasets.
> >
> > [1] Ailing Zeng, Muxi Chen, Lei Zhang, and Qiang Xu. Are transformers effective for time series
> > forecasting? AAAI, 2023.
> >
> > [2] Li, Zhe, et al. "Revisiting long-term time series forecasting: An investigation on linear mapping." arXiv preprint arXiv:2305.10721 (2023).
> >
> > [3] Toner, William, and Luke Darlow. "An analysis of linear time series forecasting models." ICML 2024
> >
> > [4] Han, Lu, Han-Jia Ye, and De-Chuan Zhan. "The capacity and robustness trade-off: Revisiting the channel independent strategy for multivariate time series forecasting." IEEE Transactions on Knowledge and Data Engineering (2024).
> >
> > [5] Nie, Yuqi, et al. "A time series is worth 64 words: Long-term forecasting with transformers." ICLR 2023
> >
> > [6] Liu, Yong, et al. "itransformer: Inverted transformers are effective for time series forecasting." ICLR 2024 spotlight
> >
> > [7] Wu, Haixu, et al. "Timesnet: Temporal 2d-variation modeling for general time series analysis." ICLR 2023
> >
> > [8] Brunton, S. L., Brunton, B. W., Proctor, J. L., & Kutz, J. N. (2016). Koopman invariant subspaces and finite linear representations of nonlinear dynamical systems for control. PloS one, 11(2), e0150171.

---

> > ### Comment · Reviewer_FaCp · 2025-05-09
> >
> > I am not sure why the authors say:
> >
> > 2. "The numerical reconstruction error is usually not discussed in the normalizing flow literature, and seems of little interest to us, which is why we do not think it deserves a detailed discussion."
> >
> > Even when you have an algebraically exact inverse, if the formula is unstable with respect to perturbations (therefore numerically ill-conditioned) it could be useless computationally. If the motivation of the authors is to claim that the invertibility of the normalizing flow is important, this seems to be a crucial issue. To be clear, many theoretically plausible invertible mappings are useless in practice because of this reason.

---

> ### Author Response · Authors · 2025-05-09
>
> We understand that some algebraically invertible expressions can be computationally useless due to numerical ill-conditioning. What we meant to say is that we do not design a new invertible function but rely on well-established coupling layer normalizing flow architectures (e.g. [1,2,3], our experiments all use [1]) which are empirically stable. The specific usage of a normalizing flow as an invertible encoder in a Koopman autoencoder is not a novelty either, but was proposed in earlier papers [4,5]. None of the papers [1,2,3,4,5] mentioned ill-conditioning issues in their experiments. We did not experiment such issues either in our own experiments, and thus, considering that the novelties of our paper do not involve new invertible functions, discussions on the numerical conditioning seem to be out of our scope. However, for a general discussion on the conditioning of coupling layer normalizing flows, one can refer to [6].
>
> [1] Dinh, Laurent, David Krueger, and Yoshua Bengio. "Nice: Non-linear independent components estimation." arXiv preprint arXiv:1410.8516 (2014).
>
> [2] Dinh, Laurent, Jascha Sohl-Dickstein, and Samy Bengio. "Density estimation using real nvp." International Conference on Learning Representations 2017
>
> [3] Kingma, Durk P., and Prafulla Dhariwal. "Glow: Generative flow with invertible 1x1 convolutions." Advances in neural information processing systems 31 (2018).
>
> [4] Meng, Yuhuang, Jianguo Huang, and Yue Qiu. "Koopman operator learning using invertible neural networks." Journal of Computational Physics 501 (2024): 112795.
>
> [5] Jin, Yuhong, et al. "Invertible Koopman network and its application in data-driven modeling for dynamic systems." Mechanical Systems and Signal Processing 200 (2023): 110604.
>
> [6] Lee, H., Pabbaraju, C., Sevekari, A. P., & Risteski, A. (2021). Universal approximation using well-conditioned normalizing flows. Advances in Neural Information Processing Systems, 34, 12700-12711.

---

> > ### Comment · Reviewer_FaCp · 2025-05-09
> >
> > If I understand correctly, the authors first asserted that the inversion is possible to machine precision (which was incorrect), and now the authors are arguing that it is "empirically stable" instead. This prompts the question: Empirically stable in what sense? Did these works test for conditioning? Or is it that these works did not face issue when testing on certain data sets? The authors say "none of the papers mentioned ill-conditioning issues in their experiments," which is even more concerning, because it points to the fact that the literature have not studied this aspect carefully. Authors say the method is "well-established", but it is also well-established that deep learning models consistently suffer from input instabilities (ill-conditioning).
> >
> > I am not arguing that the authors tackle this issue in the manuscript detail, but merely recommending that the authors point out that invertibility the authors mention is _one form_ of invertibility and does not necessarily guarantee stable and accurate invertibility. This would make it clear what kind of invertibility the authors are referring to.

---

> > > ### Author Response · Authors · 2025-05-10
> > >
> > > Following the reviewer's recommendation, we have added a footnote stating that "It should be noted that the algebraic invertibility does not guarantee stable and accurate reconstructions in practice, since normalizing flows can be subject to numerical ill-conditioning: see e.g. [6] for an extensive discussion on the conditioning of normalizing flow models." (where [6] is the paper by Lee et al. that we mentioned in our previous comment).

---

> > > > ### Comment · Reviewer_FaCp · 2025-05-11
> > > >
> > > > I thank the authors for the updates. I believe my concerns have been addressed.

---

### Decision · Action_Editor_9pwW · 2025-05-23

**Recommendation:** Accept as is

**Comment:**

This work meets the two major requirements for TMLR acceptance and (uniquely) has full support from the reviewers (who were high quality reviewers). In addition, all reviewers believe that the work is ready for publication and have had their comments fully addressed. For this reason, I recommend accept as is.

**Audience:**

All reviewers agree that this work has an audience in the TMLR community. In particular, it will be of interest to those studying time-series modeling and dynamical systems theory.

**Claims And Evidence:**

All reviewers agree that this work has sufficient and clear evidence of the underlying claims.